

**Inter-laboratory comparison of cryogenic water extraction**
**systems for stable isotope analysis of soil water**
**Natalie Orlowski[a,b*], Lutz Breuer[b,c,], Nicolas Angeli[d], Pascal Boeckx[e], Christophe**
**Brumbt[f], Craig S. Cook[g], Maren Dubbert[h], Jens Dyckmans[i], Barbora Gallagher[j],**
**Benjamin Gralher[k], Barbara Herbstritt[k], Pedro Hervé-Fernández[e,f,l], Christophe**
**Hissler[m], Paul Koeniger[n], Arnaud Legout[o], Chandelle Joan Macdonald[g], Carlos**
**Oyarzún[f], Regine Redelstein[p], Christof Seidler[q], Rolf Siegwolf[r], Christine Stumpp[s], Simon**
**Thomsen[t], Markus Weiler[k], Christiane Werner[h], and Jeffrey J. McDonnell[a]**
[a] Global Institute for Water Security, School of Environment and Sustainability, University of
Saskatchewan, Saskatoon, Canada
[b] Institute for Landscape Ecology and Resources Management (ILR), Research Centre for
BioSystems, Land Use and Nutrition (IFZ), Justus Liebig University Giessen, Giessen, Germany
[c] Centre for International Development and Environmental Research, Justus Liebig University
Giessen, Giessen, Germany
[d] INRA-UHP Ecologie et Ecophysiologie Forestières, INRA Centre de Nancy, Champenoux, France
[e] Isotope Bioscience Laboratory (ISOFYS), Faculty of Bioscience Engineering, University of Ghent,
Ghent, Belgium
[f] Instituto de Ciencias de la Tierra, Universidad Austral de Chile, Valdivia, Chile
[g] Department of Ecosystem Science and Management, Stable Isotope Facility, University of
Wyoming, Laramie, Wyoming, USA
[h] Chair of Ecosystem Physiology, University of Freiburg, Freiburg, Germany, BAYCEER; Chair of
Ecosystem Physiology, University of Bayreuth, Bayreuth, Germany
[i] Institute of Soil Science and Forest Nutrition, Centre for Stable Isotope Research and Analysis
(KOSI), University of Goettingen, Goettingen, Germany
[j] Institute for Environmental Research, Australia Nuclear Science and Technology Organization,
Sydney, Australia
[k] Faculty of Environment and Natural Resources, Chair of Hydrology, Albert-Ludwigs University
Freiburg, Freiburg, Germany
[l] Laboratory of Hydrology and Water management, Faculty of Bioscience Engineering, University of
Ghent, Ghent, Belgium
[m] Luxembourg Institute of Science and Technology (LIST), Department of Environmental Research
and Innovation (ERIN), Esch-sur-Alzette, Luxembourg, Luxembourg
[n] German Federal Institute for Geosciences and Natural Resources (BGR), Hannover, Germany
[o] INRA UR1138 Biogéochimie des Ecosystèmes Forestiers, INRA Centre de Nancy, Champenoux,
France



[p] Plant Ecology and Ecosystems Research, University of Goettingen, Goettingen, Germany
[q] Ecophysiology of Plants, Technical Universtiy Munich, Munich, Germany
[r] Stable Isotope Research Facility, Paul Scherrer Institute (PSI), Villigen, Switzerland
[s] Institute of Groundwater Ecology, German Research Center for Environmental Health, Helmholtz
Zentrum München, Neuherberg, Germany; now at: Institute of Hydraulics and Rural Water
Management (IHLW), University of Natural and Life Sciences (BOKU) Vienna, Austria
[t] Institute of Soil Science, University of Hamburg, Hamburg, Germany
*Correspondence to: N. Orlowski, now at: Faculty of Environment and Natural Resources, Chair of
Hydrology,      Albert-Ludwigs      University      Freiburg,      Freiburg,      Germany
(Natalie.Orlowski@hydrology.uni-freiburg.de)

## 12    Abstract

For more than two decades, research groups in hydrology, ecology, soil science and
biogeochemistry have performed cryogenic water extractions for the analysis of $\delta^2$H and $\delta^{18}$O
of soil water. Recent studies have shown that extraction conditions (time, temperature, and
vacuum) along with physicochemical soil properties may affect extracted soil water isotope
results. Here we present results from the first worldwide round robin laboratory
intercomparison. We test the null hypothesis that with identical soils, standards, extraction
protocols and isotope analyses, cryogenic extractions across all laboratories are identical. Two
'standard soils' with different physicochemical characteristics along with deionized reference
water of known isotopic composition, were shipped to 16 participating laboratories. Participants
oven-dried and rewetted the soils to 8% and 20% gravimetric water content, using the deionized
reference water. One batch of soil samples was extracted via pre-defined extraction conditions
(time, temperature, and vacuum) identical to all laboratories; the second batch was extracted
via conditions considered routine in the respective laboratory. All extracted water samples were
analyzed for $\delta^{18}$O and $\delta^2$H by the lead laboratory (Global Institute for Water Security, GIWS,
Saskatoon, CA) using both a laser and an isotope ratio mass spectrometer (OA-ICOS and IRMS,
respectively). We rejected the null hypothesis. Our results showed large differences in retrieved
isotopic signatures among participating laboratories linked to soil type and soil water content
with mean differences to the reference water ranging from +18.1‰ to –108.4‰ for $\delta^2$H and
+11.8‰ to –14.9‰ for $\delta^{18}$O across all laboratories. In addition, differences were observed
between OA-ICOS and IRMS isotope data. These were related to spectral interferences during
OA-ICOS analysis that are especially problematic for the clayey loam soils used. While the



types of cryogenic extraction lab construction varied from manifold systems to single chambers,
no clear trends between system construction, applied extraction conditions, and extraction
results were found. Rather, differences between isotope results were influenced by interactions
between multiple factors (soil type and properties, soil water content, system setup, extraction
efficiency, extraction system leaks, and each lab's internal accuracy). Our results question the
usefulness of cryogenic extraction as a standard for water extraction since results are not
comparable across laboratories. This suggests that defining any sort of standard extraction
procedure applicable across laboratories is challenging. Laboratories might have to establish
calibration functions for their specific extraction system for each natural soil type, individually.

## 1 Introduction

The interpretation of the stable isotope signatures of water ($\delta^2$H and $\delta^{18}$O) from soils in many
research disciplines relies on accurate, high-precision measurements (Wassenaar et al., 2012).
To extract water from soils for isotopic analysis, cryogenic water extraction (CWE) is the most
widely used laboratory-based removal technique (Araguás-Araguás et al., 1995; Orlowski et
al., 2016a). The ability to obtain measurable amounts of water from small sample sizes
(i.e. < 10 g) makes this method attractive. However, CWE is also accompanied by high capital
and operating costs. Despite its widespread use, recent work has identified several extraction
artifacts leading to uncertain isotopic signature identification (Gaj et al., 2017a; Orlowski et al.,
2016b). Studies have shown that extraction conditions (i.e., extraction time, temperature, and
vacuum) need to be adapted specifically to the soil used (Araguás-Araguás et al., 1995; Gaj et
al., 2017a; Meißner et al., 2014; Orlowski et al., 2016a). Notwithstanding, isotope effects
triggered by physicochemical soil properties (e.g., clay minerals, soil organic carbon content,
and water content) can occur (Araguás-Araguás et al., 1995; Gaj et al., 2017a; Meißner et al.,
2014; Oerter et al., 2014; Orlowski et al., 2013). However, the ecohydrology and soil science
communities currently lack clear recommendations for standardized water extraction conditions
from soils. Although there seems to be an agreement on the need to control the extraction yield
of cryogenic extraction facilities (recovery rate in percentage of previously added water), there
exists a large variability in the applied extraction conditions between laboratories. Moreover,
extraction systems vary in terms of heating elements, size of extraction containers, or
throughput, in addition to the aforementioned extraction conditions (Goebel and Lascano, 2012;



Hydrology and
Koeniger et al., 2011; Orlowski et al., 2013). Thus, no standard system setup or methodology
exists.
Despite the work to date and the extensive application of stable water isotope analysis, no
formal interlaboratory comparison between different cryogenic systems has been published.
Here we present the first worldwide interlaboratory comparison between 16 different cryogenic
extraction facilities. CWE procedures were conducted with two standard soils with different
physicochemical characteristics (silty sand and clayey loam), spiked with a known isotopic
label at different gravimetric water contents (WC of 8% and 20%). The null hypothesis guiding
this work was that all laboratories would yield the same results independent of soil type and
water content. In addition, we addressed the following research questions:
1. How does the cryogenic system configuration affect resulting soil water isotopic

12       composition?

2. How do soil type and soil water content affect the isotope results?
3. How do results differ when extracted soil water stable isotopic compositions are

15       measured via off-axis integrated cavity output spectroscopy (OA-ICOS) vs. isotope

16       ratio mass spectrometry (IRMS)?

4. What do we learn from this exercise for standardization of cryogenic extraction

18       facilities?

**2 Methods**
**2.1 Experimental design**
Table 1 provides a description of the respective extraction systems that participated in the
intercomparison. In total, 16 independent laboratories from seven countries took part in the
trial.

26                         [Table 1 near here]

Before the commencement of the round robin test, participants were asked to fill out a
questionnaire (see Appendix 1) to characterize their cryogenic extraction system in terms of
numbers of extraction slots or amount of sample material usually introduced into the system
(size of extraction unit). Two standard soils with different physicochemical properties (clayey





loam and silty sand) from the German State Research Institute for Agriculture (LUFA Speyer,
2015) (Table 2) were used for the interlaboratory comparison.
[Table 2 near here]
We chose a silty sand from which we expected water extractions to be relatively easy for each
laboratory without cation ion exchange problems, and a clayey loam soil,—which is known to
be challenging for CWE extraction systems. Clayey soils can be difficult due to interactions
with the clay fraction and different types of clay minerals—the so-called adsorbed cation effect
(Oerter et al., 2014). Clay soils also present challenges with regard to the tightness of water
bound to mineral surfaces which causes an additional isotope effect (Ingraham and Shadel,
1992; Oerter et al., 2014; Walker et al., 1994).
Soil samples were sieved to a grain size <2 mm. Soils were pre-dried at 105°C for 48 h,
homogenized, and shipped in tightly sealed glass bottles to the 16 independent laboratories
along with deionized (DI) reference water of known isotopic composition (measured on both
an IWA-45EP Analyzer (OA-ICOS, Los Gatos Research Inc., Mountain View, US): $\delta^2$H:
−59.8±0.2‰ and $\delta^{18}$O: −8.5±0.1‰, n=6; and via Delta V™ Advantage mass spectrometer
(Thermo Fisher Scientific, Waltham, MA, US): $\delta^2$H: −60.5±0.2‰ and $\delta^{18}$O: −8.7±0.1‰, n=6).
All bottles containing either soils or DI water were filled, capped tightly, and wrapped with
Parafilm® to prevent water loss. We decided not to ship ready-to-use rehydrated soils to avoid
evaporation fractionation effects and to give participants the opportunity to adjust e.g. samples
sizes to the specific requirements of their extraction system. Water loss and evaporative
enrichment from the shipped DI water was checked by isotopic comparison of shipped and non-
shipped DI water (1. shipment test: Giessen–Freiburg (Germany)–Saskatoon (Canada) and 2.
shipment test: Giessen (Germany)–Saskatoon (Canada) vs. non-shipped water samples). After
this simple experiment, isotope fractionation effects due to shipment were excluded.
As a reliability test, each participant in the intercomparison performed water-water cryogenic
extractions (defined here as simply extracting pure water, i.e. without any soil material present)
using their extraction facility. This was done in order to determine the capability of the
respective extraction apparatus to recapture water of known isotopic composition. After
showing the operational reliability, CWEs with the rehydrated soil samples were performed
following a pre-defined protocol.



## 2.2 Sample preparation protocol

Before starting the rewetting of the pre-dried soil samples with the DI water, participants oven-dried (at 105°C for 48 h) the provided soils again to remove any potential water that could be present (e.g., remoistening of the soil samples during shipment). Afterwards, soils were placed in a desiccator for cooling and to prevent remoistening of the dried soil samples with ambient water vapor (Orlowski et al., 2016b; Van De Velde and Bowen, 2013). For rehydration, two different amounts of reference DI water were added to the respective soil types (to create 8% and 20% gravimetric WC). Exposure of the dried soil samples to ambient conditions was kept as short as possible. Participants adjusted the amount of respective soil material and water for rewetting the samples according to the specific requirements of their extraction system e.g., size of extraction containers. Sample preparation was performed separately for OA-ICOS and IRMS analysis but in the identical way as specified below:

1. Soil and DI water were added in an alternating fashion. A quarter of soil material (clayey loam/silty sand) and a quarter of DI water were alternatively added to the pre-weighed extraction tube to facilitate soil-water-homogenization.

2. This rewetting procedure was completed by adding a quarter of soil material to the extraction tube to avoid supernatant water and to obtain the best possible mixing.

3. Samples were weighed again.

4. Finally, an inert cover (Fackelmann Inc, Hersbruck, DE) was placed on top of the soil sample to avoid the spread of sample material throughout the respective cryogenic extraction line. The inert material was proven to not cause isotope effects during soil water extraction (Orlowski et al., 2013).

5. Extraction tubes were plugged and sealed with Parafilm® to ensure an air-tight system.

6. Rehydrated soils in their respective extraction containers were placed in vertical position in a refrigerator (5°C for 72 h), which further allowed the liquid and solid phase to equilibrate.

## 2.3 Cryogenic extraction approaches

Each laboratory was instructed to follow two different extraction approaches: (I) For the first subset of rehydrated soil samples, participants applied the CWE procedure considered routine in their laboratory for the specific soil type and soil water content. (II) With the second subset,

CWE under pre-defined conditions for all labs was performed: For silty sand, a 45 min
extraction time was used while 240 min was applied to clayey loam samples, both at an
extraction temperature of 100°C and a vacuum of 0.3 Pa. These pre-defined extraction
parameters were identical for all participating laboratories. Three replicates per soil type and
soil water content resulting in 24 samples per extraction procedure (pre-defined and laboratory
specific) and isotope analysis method (OA-ICOS and IRMS) were processed (n=48 in total).
Pre- and post-oven-dried (105°C for 24 h) soil sample weights were used to determine water
recovery rates. All extracted water samples were transferred to 2 mL amber glass vials capped
with solid lids (Th. Geyer Inc., Renningen, DE), tightly sealed with Parafilm®, labeled, and
shipped to the GIWS for isotope analysis. If the amount of extracted water was not sufficient
to entirely fill the 2 mL vial, inserts (0.2 mL) were used (Th. Geyer Inc., Renningen, DE) to
minimize sample vial headspace, following standard procedures as outlined by the IAEA

13 (2014).

**2.4 Isotope analyses**
For cross-checking isotope results and ruling out potential lab analytical differences, the
isotopic composition of the extracted water samples was analyzed via both OA-ICOS and
IRMS. OA-ICOS samples were analyzed on an IWA-45EP Analyzer (Los Gatos Research Inc.,
Mountain View, US). The accuracy of OA-ICOS analyses was ±0.5‰ for $\delta^2H$ and ±0.1‰ for
$\delta^{18}O$. IRMS samples were analyzed on a Delta V™ Advantage mass spectrometer (Thermo
Fisher Scientific, Waltham, MA, US) and an H/Device peripheral using a Cr-reduction method
for $\delta^2H$ analysis (Morrison et al., 2001). For $\delta^{18}O$ analysis, a GasBench II peripheral was
utilized. Using mass spectrometry, a conversion from the water into a light gas suitable for mass
spectrometry ($H_2$, $CO_2$, $CO$, $O_2$) is necessary. This conversion step often turns out to limit the
achievable precision of IRMS (Brand et al., 2009). In our case, IRMS results are accurate to
±1‰ for $\delta^2H$ and to ±0.2‰ for $\delta^{18}O$, respectively. All isotope ratios are reported in per mil (‰)
relative to Vienna Standard Mean Ocean Water (VSMOW) ($\delta^2H$ or
$\delta^{18}O=(R_{sample}/R_{standard}-1)\times1000‰$), where R is the isotope ratio of the sample and the known
reference (i.e. VSMOW)) (Craig, 1961). In-house standards, calibrated against VSMOW2 and
SLAP2, were run as samples to allow the results to be reported against VSMOW (Nelson,

31 2000).





OA-ICOS isotope data of soil water extracts were checked but not corrected for spectral
interferences (caused by potentially co-extracted organics such as methanol or ethanol) using
the Spectral Contamination Identifier post-processing software (LWIA-SCI, Los Gatos
Research Inc.) when measured via OA-ICOS. This software compares recorded spectra from
unknown samples with those from known non-contaminated samples (standards) to produce a
metric of contamination from either narrow-band (e.g., methanol (MeOH)) or broad-band (e.g.,
ethanol (EtOH)) absorbers which indicates the likelihood or degree of spectral interference
(Schultz et al., 2011). IRMS results are generally not affected by organic contaminants.
**2.5 Statistical evaluation**
We used R for statistical analyses (R version 3.3.2; R Core Team, 2014). For quantifying
laboratory variances, differences between pre-defined and laboratory specific extraction
procedures, effects of soil type and WC, differences between OA-ICOS and IRMS, all data
were tested for normality using the Shapiro-Wilk test. Homoscedasticity was tested using either
the Levene's test for normally distributed data or the Fligner-Killeen test for non-normally
distributed data. Cook's distance was determined in order to identify outliers ($D>1$). Depending
on the type of data (normally distributed and homoscedastic), either Kruskal-Wallis rank sum
tests or Analyses of Variances (ANOVAs) were applied and posthoc tests (e.g., Nemenyi-tests)
were run to determine which groups were significantly different ($p \leq 0.05$). P-value adjustments
via the FDR-method (false discovery rate) were applied to reduce the family-wise-error rate
(Zieffler et al., 2012).
For graphical comparisons, a target standard deviation (TSD) for acceptable performance was
set to ±2‰ for $\delta^2H$ and ±0.2‰ for $\delta^{18}O$ similar to Orlowski et al. (2016b), which is considered
reasonable for hydrologic studies (Wassenaar et al., 2012). The TSD does not account for errors
associated with the extraction method itself, weighing errors, and volumetric water additions to
the sample, or any standard deviations (1SDs) related to the isotope analysis. Statistically
significant ($p \leq 0.05$) linear regressions were added to dual isotope plots as references
(evaporation water lines) as well as the Global Meteoric Water Line (GMWL: $\delta^2H = 8.2 \times \delta^{18}O$
$+ 11.3‰$, as defined by Rozanski et al. (1993)).




**3 Results**
**3.1 Cryogenic extraction systems and water extraction efficiencies**
Cryogenic extraction systems varied greatly from lab to lab: from manifold, high-throughput
devices (as described by Orlowski et al. (2013)) to small, single chamber systems (as in
Koeniger et al. (2011) and West et al. (2006)) (for details see Table 1). The systems showed
differences in terms of the extraction containers (form, size, volume, and material), the heating
module and its application temperature (heating tapes or lamps, water baths or hot plates), the
type of fittings and connections (glass, stainless steel), as well as in the vacuum producing units
(Table 1).
To determine the degree of extraction efficiency for each lab's samples, water recovery rates
were calculated for those labs that provided the complete set of soil weight data (in % of
previously added water). When comparing water recovery rates against isotope results, the
clayey soil showed no clear trend (Fig. 1). Even if water recovery rates were higher than 98%
(following the definition of Araguás-Araguás et al. (1995)), extracted isotope results differed
from the reference DI water (Fig. 1). For example, at 8% soil water content (WC), recovery
rates of above 98% were achieved, but isotope results were depleted in comparison to the
reference DI water (Fig. 1, left panels).

19                               [Figure 1 near here]

For the silty sand, recovery rates were generally higher in comparison to the clayey soil. Only
a few samples showed extraction efficiencies lower than 98% (Fig. 1, right panels).
Surprisingly, we observed some recovery rates higher than 100%. This was especially an issue
for soils at 8% WC (Fig. 1).
Correlation analysis was performed in order to relate extraction parameters (i.e., time,
temperature or vacuum) to OA-ICOS and IRMS isotope results.

28                               [Figure 2 near here]

We found no significant correlations between the extraction parameters and the respective
isotope results, exemplarily shown for $\delta^2H$ results (Fig. 2) (e.g., $R^2$=0.0 for $\delta^2H$ vs. duration or
temperature).





**3.2 Laboratory performance with respect to water content and soil type**

Figures 3 and 4 show the mean differences between the extracted samples via the lab procedure's extraction approach (I) and the pre-defined extraction approach (II) and the reference DI water $\delta^2$H and $\delta^{18}$O values, respectively.

For the 8% WC tests, mean differences for the clayey loam ranged from +13.1 to −32.8‰ for $\delta^2$H. For the individual lab procedure's extraction approach (I) at 8% WC for the clayey loam, two laboratories (lab 3 and 8) were able to get back to the reference $\delta^2$H value based on no statistically significant differences ($p > 0.05$) (Fig. 3, upper left plot). For the pre-defined extraction approach (II) at 8% WC, two other labs recovered the $\delta^2$H value from the clayey loam (lab 9 and 15).

For soil samples with 20% WC, variation among laboratories was smaller but only one laboratory (lab 9) recovered the reference DI water $\delta^2$H value applying the pre-defined extraction approach for the clayey loam. Mean differences between the clayey loam extracts and the reference DI water ranged from +2.8 to −19.5‰ (Fig. 3, upper right plot).

[Figure 3 near here]

Mean differences between the silty sand water extraction and the reference $\delta^2$H signature were in a smaller range of ±18‰ than clayey loam extracts from the same treatment (8% WC). For the individual lab procedure's extraction approach (I) at 8% WC, five laboratories recovered the added label from the silty sand (Fig. 3, lower left plot) with no statistical differences between the reference DI water ($p > 0.05$) (labs 6, 8, 9, 13, and 15), whereas for the pre-defined extraction approach (II) at 8% WC, three labs got back to the added $\delta^2$H value (labs 9, 12, and 15).

For silty sand at 20% WC, most laboratories' results even fell close to the range of the TSD of ±2‰. Mean differences to the reference DI water $\delta^2$H signature ranged from +8.5 to −15.1‰ (Fig. 3, lower right plot). However, extraction approach I was statistically not successful in recovering the added label ($p < 0.05$), but five laboratories (6, 9, 10, 14, and 15) showed no significant differences to the reference DI water when applying extraction approach II to the silty sand at 20% WC.

Laboratories performed better for $\delta^{18}$O signature recovery, especially with extraction approach I. For both clayey loam WC treatments, labs 13 and 15 were the most successful. Again, mean



differences to the reference DI water were larger for the 8% WC than for the 20% WC (Fig. 4,
upper plots). However, for the clayey loam at 20% WC with the pre-defined approach (II) only
lab 13 and 14 did not show statistically significant differences to the added $\delta^{18}O$ signature
(Figure 4, upper right plot) (p>0.05).
[Figure 4 near here]
For the silty sand, most laboratories were able to get back the known value with no statistically
significant differences to the reference $\delta^{18}O$ value (Fig. 4, lower plots). For both WC treatments
of the silty sand, extraction approach II seemed to work better in recovering the added label.
Across both soil types, WC treatments, and extraction approaches, lab 13 was the most
successful in recovering the reference $\delta^{18}O$ value, whereas for $\delta^2H$ recovery lab 9 gained back
the added label in most of the cases.
In general, isotope results were neither comparable between laboratories nor between one
laboratory at different soil types or WCs, meaning that a specific laboratory, for example,
successfully recovered the added DI water value for silty sand but was not able to gain back the
known label for clayey loam. Moreover, recovery results differed between both isotopes. For
example, lab 13 was the most successful for $\delta^{18}O$ but not for $\delta^2H$ signature recovery. In terms
of lab internal reproducibility, some labs showed small standard deviations for the replicates of
the same soil type at a given WC (Fig. 3 and 4); even so, isotope results differed statistically
significant from the introduced reference DI water.
**3.3 Differences between OA-ICOS- and IRMS-based measurements**
Figures 5 (clayey loam) and 6 (silty sand) illustrate data variability for each laboratory and WC
with respect to the labeled reference DI water added to each soil type in dual isotope space.
Significant differences were observed between OA-ICOS and IRMS isotope data sets (p≤0.05).
The clay soil isotope data at 8% WC showed the greatest differences between OA-ICOS and
IRMS measurements (mean differences of 1.3 and 1.2 for $\delta^2H$ and $\delta^{18}O$, respectively). Smallest
differences between isotope analyzers were observed between both WC treatments of the silty
sand (Fig. 6). The data sets with the lowest SD for both isotopes across labs and extraction
approaches were the silty sand samples at 20% WC measured via OA-ICOS and IRMS (SD of

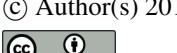


±3.1 for $\delta^2$H measured via OA-ICOS and ±4.2 for IRMS, respectively). However, those data
sets still did not reach the TSD of ±2‰ for $\delta^2$H and ±0.2‰ for $\delta^{18}$O.

4                                [Figure 5 near here]

For comparison, apart from evaporation lines, the GMWL is also given in each subplot.
Interestingly, isotope data across laboratories plot on slopes lower than the GMWL. For both
soil types, evaporation lines of the IRMS measurements showed better correlations (for the silty
sand $R^2$=0.8 and 0.9 for 8% and 20% WC, respectively) than those of OA-ICOS measurements
($R^2$=0.7 for 8% and 20% WC) (Fig. 6). Silty sand's soil evaporation water lines showed greater
slopes (5.4–7.2 across both WCs and isotope analysis) than clayey loam's soil evaporation
water lines (2.8–5.2 across both WCs and isotope analysis) (Figures 5 and 6). The clayey loam
evaporation lines for the higher WC also showed greater slopes than those of the lower WC
(Fig. 5). Isotopic fractionation due to evaporation leads to a stronger kinetic effect for $^{18}$O
compared to $^2$H, resulting in evaporative enrichment of the water along an evaporation water
line (e.g., soil evaporation water line) with a lower slope relative to the original water
(Gonfiantini, 1986) –in our case, the reference DI water.
For the clay soil type, the IRMS data sets (8% and 20% WC) plot closer to the GMWL and the
analyzed values showed a smaller SD in comparison to the OA-ICOS assays (SD of ±8.4 for
the OA-ICOS $\delta^2$H data vs. ±7.5 for the OA-ICOS data at 8% WC) (Fig. 6).

22                                [Figure 6 near here]

In general, the spread of the isotope data decreased from 8% to 20% WC and from OA-ICOS
to IRMS measurement results (Figures 5 and 6). The OA-ICOS isotope analyses showed more
outliers than those of IRMS. Moreover, fewer outliers were found among the silty sand data
when compared to that of the clayey loam soil. Overall, IRMS results for all soil types and WCs
were slightly more depleted than those of OA-ICOS. However, differences were not significant
(p>0.05). In general, most of the water extracts were depleted in comparison to the reference
DI water, which is especially true for $\delta^2$H.
Examination of the differences between OA-ICOS and IRMS data, prompted to test the OA-
ICOS data for spectral interferences. Figure 7 shows that for the clayey loam soil, differences





between OA-ICOS and IRMS data might be due to co-extracted alcoholic compounds, which
caused erroneous OA-ICOS data.

4                                    [Figure 7 near here]

Few samples among the 8% WC versions of clay water extracts showed issues with both broad-
band and narrow-band absorbers. This contamination by both methanol and ethanol explained
the outliers found at 8% WC in the clayey loam data (Fig. 7, upper left plot). Among these data,
only a small number of samples showed no contamination, which were interestingly more
depleted in comparison to data flagged as affected by narrow-band absorbers. For the silty sand
soil, only a few samples were contaminated and flagged as affected by narrow-band absorbers.
Interestingly, outliers in the silty sand soil data set at 8% WC could not be explained by narrow-
or broad-band absorbers.
**4 Discussion**
**4.1 Why are the cryogenic extraction results different across the participating laboratories?**
We rejected our null hypothesis that all laboratories would yield the same results independent
of soil type and water content. We showed that cryogenic extraction results were not
comparable among laboratories. We also observed differences in the ability of individual labs
to recover both isotope values ($\delta^2$H and $\delta^{18}$O) of the added reference DI water. Some
laboratories were able to get back to the reference $\delta^2$H value but were not successful for $\delta^{18}$O.
Each extraction system setups were different. Therefore, it was difficult to give any
recommendation with regard to a high-performance and accurate extraction system that would
lead to overall successful extractions. As a quality control, we checked water recovery rates,
which were in some cases even higher than 100% (Fig. 1). This could be attributed either to
leaky vacuum systems (which might allow atmospheric water vapor to enter the system) or to
a remoistening of the oven-dried soil samples before water extraction. Remoistening of oven-
dried soil samples might be a general problem of such spiking experiments. In our case, sample
preparation was not performed under an inert gas flow and, unfortunately, data on temperature
and relative humidity conditions under which sample preparation took place are unavailable
from the respective labs. Ambient water vapor isotopic composition measurements would have



also been a relevant additional information. Contamination could also occur when an extraction
system is not dried or cleaned after each extraction run, leaving moisture and/or soil material
behind which would affect the next sample's results. Other measurement uncertainties during
the extraction protocol could arise from weighing errors (scale calibration and precision), the
accuracy of the volume of water additions to the soil samples, transfer of the samples, loss of
water vapor during evacuation of the extraction system, unsteady heating temperatures,
condensation of water vapor in the extraction system, and a lack of precision of analytical and
laboratory equipment.
It is also possible that participating labs did not follow the pre-defined extraction procedure (II)
in the exactly same ways. Even extraction results from some individual labs for the same soil
type and WC showed high SDs (Fig. 3 and 4) which questions the overall repeatability of
individual water extraction results. For the first, "in-house" extraction approach (I), not all
laboratories indicated the precise extraction conditions (extraction temperature, time, and
vacuum) that they used for the specific soil types and WCs.
As an additional performance test, laboratories were asked to perform water-to-water
extractions to show their ability to recover water of known isotopic composition prior to soil-
based tests. For example, some laboratories, like lab 2, showed a high accuracy for these water-
to-water extractions of $\pm0.4‰$ for $\delta^2H$ and $\pm0.1‰$ for $\delta^{18}O$ (n=119) as well as lab 16. They
performed extraction tests with tap water, which resulted in no significant differences between
the initial, untreated ($-56.7‰ \pm 0.4$ for $\delta^2H$ and $-9.3‰ \pm 0.1$ for $\delta^{18}O$) and extracted tap water
($-57.5‰ \pm 0.6$ for $\delta^2H$ and $-9.4‰ \pm 0.1$ for $\delta^{18}O$). These examples show that these labs among
others were able to reach the TSD with simple water-to-water extractions, but with soils, they
were unsuccessful. This indicates that differences between the reference DI water and water
spiked and extracted from soils are likely caused by interactions with soil particles.
Given our findings, we now question the standard quality controls (e.g., water recovery rate
calculations and water-to-water extractions). Quality controls with spiked soil samples may be
a more effective way to demonstrate lab's internal accuracy. However, such spiking
experiments as performed in our study come along with other issues as recently outlined by Gaj
et al. (2017b) and Sprenger et al. (2015). Gaj et al. (2017a) applied the Rayleigh equation (using
stable isotope signatures) to calculate how much water was cryogenically extracted from pure
clay minerals. They found that for samples from which water has been extracted to 100%





(determined gravimetrically), the Rayleigh equation showed that only 72% of water was
extracted at a temperature of 105°C. When using an extraction temperature of 205°C, the
Rayleigh-estimated amount of water extracted was close to 90%, but still not 100%. This result
clearly shows that despite the gravimetric quality control suggesting that all water has been
extracted, isotopic differences may still exist.
Overall, laboratories 9 (for $\delta^2H$) and 13 (for $\delta^{18}O$) were the most successful in getting back to
the DI reference water over all soil types and WCs. For the lab's in-house procedure, laboratory
9 extracted both soils for 90 min at 95°C and 0.8 Pa. Their reported water extraction efficiency
was 99–100%. Glass tubes were used as extraction containers and a water bath as heating
element. Laboratory 13 used different extraction parameters, which also varied slightly from
sample to sample: for the clayey loam at 8% WC, extractions were conducted for 75–114 min
at 150–100°C and 8–13.3 Pa. For the 20% WC, they used 266 min at 100°C and 6.7–13.3 Pa
as in-house extraction parameters. For the silty sand at 8% WC, their extraction time was 15 min
at 100°C and 7.3–13.3 Pa. For the 20% WC, they extracted for 30 min at 100°C and 6.7–
10.7 Pa. Lab 13 further specified that their extraction times were dictated by a decline in the
pressure level indicating that no more water was evaporating from the respective sample.
Extraction efficiencies for lab 13 varied between 93–127 %. Glass tubes were used as extraction
containers along with a sensor-regulated tube-shaped heating element. This example shows that
even for the relatively successful laboratories, extraction parameters did not seem to play a
major role for achieving the reference DI water isotopic signature.
**4.2 How do soil type and water content affect the results?**
The adsorbed and interlayer water occurring in clayey soils can complicate the interpretation
of obtained isotope data. Clay-water sorption capacity is well known (Schuttlefield et al., 2007;
White and Pichler, 1959). White and Pichler (1959) found early on that montmorillonite adsorbs
more water than kaolinite, illite, and chlorite, while chlorites and illites have similar water-
sorption properties. The amount of water absorbed/adsorbed by clay minerals ranges from 800-
500% for Na-montmorillonite (Kaufhold and Dohrmann, 2008; White and Pichler, 1959) to as
low as 60% of the initial dry weight for biotite (White and Pichler, 1959). The clayey loam in
our study was a vermiculite-rich (43 relative %) 2:1 clay type, while the silty sand had a
negligible clay-fraction (2.6%) where illite (2:1 clay type) occurred with 28 relative % (Table

32 2).





Since Grim and Bradley (1940), we know that the absorbed/adsorbed water is difficult to
remove. Savin and Epstein (1970) as well as Van De Velde and Bowen (2013) have
demonstrated that the removal of interlayer and adsorbed water on clay soils can occur when
they are heated at 100 to 300°C under vacuum conditions. After clay minerals lose all their
water, their structure changes. Hence, care should be taken in order to remove clay minerals'
water, but keeping their structure. Otherwise, rewetting experiments as presented here in our
intercomparison might not be valid.
Savin and Epstein (1970) also observed that atmospheric vapor exchanged isotopically with
interlayer water (almost completely) and Aggarwal et al. (2004) showed that this can occur
within hours. This demonstrated that the isotopic composition of clay interlayer and adsorbed
water can reflect the isotopic composition of atmospheric water vapor at the storage location.
However, once the soil has been heated under vacuum and the interlayer water removed, the
remaining water showed no evidence of isotopic exchange. Again, it should be stressed here
that for our intercomparison soil samples were oven-dried twice (before and after shipment)
prior to any rewetting and labs were advised to store the dried samples in a desiccation chamber
until use. However, oven-drying was performed at an intermediate temperature (105°C for 48h)
and not under vacuum as per Savin and Epstein (1970) and different indoor laboratory 'climatic
conditions' at the participating laboratories were observed. Thus, it might be possible that not
all of the clay interlayer and adsorbed water was removed or made isotopically non-
exchangeable, and that non-equilibrium isotopic fractionation occurring at different
temperatures during heating might be responsible for some of the differences we observed.
Thus, sample preparation might have played its role, when it comes to discrepancies in lab's
results, especially those at low water contents. At these low water contents, the available water
fraction is small and exchange with interlayer and adsorbed water would be proportionally
higher. In hindsight, repeating this work with soils dried under vacuum and at higher
temperatures (i.e., 300 °C following Savin and Epstein (1970)) may help to clarify and to isolate
the effect of remaining water in clay minerals. However, so far, regular oven-drying of soils is
standard practice (Koeniger et al., 2011) for such rewetting experiments in the literature.
We also observed water content effects on the recovered isotope data as per Meißner et al.
(2014). Isotope results across labs were closer to the added reference water isotopic composition
at higher WCs. However, this isotope effect cannot be considered independent from other soil
property effects such as clay mineral water interactions or effects caused by cation exchange





capacity (CEC). Oerter et al. (2014) demonstrated that isotope effects due to soil type are more
common in soils with high cation exchange capacity (CEC) at low WCs. This can be further
exacerbated by the cations present in the soil. Those soils with high ionic potential (e.g., $Ca^{2+}$
and $Mg^{2+}$) can create large amounts of structured water surrounding them (hydrated radii)
compared to the bulk water in the system. From an oxygen isotope perspective, O'Neil and
Truesdell (1991) showed that those cations are capable of causing fractionation between bound
and bulk soil water. Moreover, soils higher in potassium ions may have a greater effect on
hydrogen isotopes, while sodium soils demonstrate non-fractionating effects (Oerter et al.,
2014). These cation fractionation effects for montmorillic soils, in particular, can result in a
depletion of up to 1.55‰ in dry soils and 0.49‰ for $\delta^{18}O$ for wet soils. In our study, chemical
and salinity effects –which occur due to the fractionation of water molecules into hydration
spheres around fully solvated cations compared to the pure water used to make the solutions –
can be ignored for the silty sand due to a low CEC of 4.1 cmol(+) $kg^{-1}$. The high CEC
(30.6 cmol(+) $kg^{-1}$) of the clayey loam soil may have caused some of the detrimental effects
seen across laboratories. This is especially the case for low WCs due to ion hydration effects
among the cations present (Table 2).

Gaj et al. (2017a) found out that the higher the abundance of $Al_2O_3$ or $Fe_2O_3$, commonly found
in clay rich soils, the lower the ability to isotopically recover added water during spiking
experiments. Our clayey loam contained 65% of $SiO_2$, but still 9% of $Al_2O_3$, which might have
affected the obtained isotope results in general but cannot be an explanation for the high
variability across labs.

Moreover, for environmental studies, the plant available water is of interest, which is not
necessarily the same than the extracted water (Orlowski et al., 2018).

### 4.3 Differences between OA-ICOS- and IRMS-based measurements

Our OA-ICOS vs. IRMS comparison showed that isotope results were significantly different
between the two isotope measurement methods.

Others have found differences in isotope results obtained from OA-ICOS and IRMS (Martín-
Gómez et al., 2015; Wassenaar et al., 2012). In a recently performed test, 235 international
laboratories conducting water isotope analyses by OA-ICOS and IRMS were evaluated.
Wassenaar et al. (2018) could show that inaccuracy or imprecise performance stemmed mainly
from skill- and knowledge-based errors including: calculation mistakes, inappropriate or



compromised laboratory calibration standards, poorly performing instrumentation, lack of
vigilance to contamination, or inattention to unreasonable isotopic outcomes. For the analysis
of $\delta^{18}O$ and $\delta^2H$ via OA-ICOS, Penna et al. (2012) showed that between-sample memory effects
can be an additional problem. Memory effect ranged from 14% and 9% for $\delta^{18}O$ and $\delta^2H$
measurements, respectively, but declined to 0.1% and 0.3% when the first ten injections of each
sample were discarded.
An additional source of error in our study might be that sample preparation for water extraction
was performed separately for OA-ICOS and IRMS analysis, but labs were instructed to follow
the exact same procedure. Nevertheless, extractions were performed on independent samples,
which might have led to differences in the extracts' isotope results.
Leen et al. (2012) and West et al. (2010) have observed effects of co-extracted organic
compounds leading to sample contamination. This can have a knock-on effect on isotope
measurements via OA-ICOS. In our study, we found effects caused by organic contamination
producing spectral interferences during OA-ICOS measurements (Fig. 7). This was mainly a
problem for the clay soil water extracts, where we found narrow- and broad-band absorbers to
be responsible for some of the outliers in the data sets. It did not seem to be a major issue for
the silty sand soil water extracts. However, some labs applied longer extraction times to the
clayey loam samples (see Fig. 2) which might have favored the co-extraction of organics.
During an intercomparison water recovery experiment, Walker et al. (1994) faced difficulties
to retrieve the added reference water from dry and wet clays, sand, and gypseous sand. They
assumed that decomposition of organic matter or extraction of clay structural water could have
caused isotope effects. Recently, Orlowski et al. (2016a) observed that $\delta^2H$ values correlated
significantly, and became progressively lighter with increasing organic carbon content when
using CWE. In environmental organic matter, the different existing exchangeable (i.e. labile)
hydrogen fractions (O-, N-, and S-bonded or aromatic hydrogen) can easily interact with
ambient water or water vapor (Ruppenthal et al., 2010) and thus are assumed to be the cause of
the isotope effects.
Nevertheless, the less expensive, rapid option of the OA-ICOS is still a viable alternative for
routine isotope analyses if no organic contamination issues are found and six or more injections
are performed and the first two or more are discarded (Penna et al., 2012). If organics are
present, proper correction schemes as per Martín-Gómez et al. (2015) need to be applied,
especially when OA-ICOS data is used in ecohydrological studies. However, so far, correction





procedures only account for contamination caused by methanol or ethanol but plant and soil
water extracts can contain a variety of different contaminants. Our work showed that the silty
sand soil water extracts were mainly free from organic contamination (Fig. 7). Still, data post-
processing is highly recommended to detect issues occurring from co-extracted alcoholic
compounds.
**4.4 Take home messages about cryogenic water extraction**
Our lab intercomparison did not find significant correlations between extraction condition
parameters such as temperature, time, and applied vacuum, and the obtained isotope results
(Fig. 2). Others have shown that extraction time and temperature have significant effects on the
CWE isotope results (Goebel and Lascano, 2012; Koeniger et al., 2011; Orlowski et al., 2013,
2016a; West et al., 2006). Gaj et al. (2017b) showed clear relationships between temperature
and the release of water from interlayer cations and organics during CWE, which affected
isotope results. They suggested using temperatures between 200°C and 300°C for clay water
extractions. However, higher temperatures could cause a release of water by oxidation of
organics and dihydroxylation of hydroxide-containing minerals, and this water might not be an
ecohydrologically active part in the water cycle. Moreover, the co-extraction of organics could
become more important at harsher extraction conditions leading to spectral interferences when
OA-ICOS is used. Our interlab comparison was not able to provide any recommendations with
regard to higher temperatures or longer extraction times leading to possibly better extraction
results. Little is known about how the applied extraction pressure affects the CWE isotope
results. But one thing is clear: that CWE is a 'brute force technique' (Orlowski et al., 2016a) in
the sense that it is not able to distinguish between waters held at different soil tensions being of
different importance for the ecohydrological water cycle. New instrumentation to sample
discretely along the moisture release curve is desperately needed (McDonnell, 2014).
We found significant differences between extraction approach I (lab "in-house" procedure) and
II (pre-defined extraction parameters). Both approaches showed significant differences to the
added reference water for the OA-ICOS results, but in different ways. For example, for $\delta^2$H
signature recovery from silty sand, extraction approach II worked better. The same was true for
$\delta^{18}$O signature recovery for both WC treatments. However, for other settings, it was difficult to
identify the ideal extraction approach that got closer to the reference DI water isotopic





composition. We found no clear tendency for which approach should be applied, thus at present,
and much to our dismay, we cannot define any standard protocol for CWE.
We could show with our interlab comparison that a number of factors affect CWE results among
which soil properties such as clay mineral composition and concomitant release of interlayer
water seemed to be important. It is therefore essential to obtain detailed soil property
information to be able to apply post-corrections as per Gaj et al. (2017a). Further research is
urgently needed to analyze the full extent of soil organic matter effects (i.e. exchangeable
bonded hydrogen (Meißner et al., 2014)) in organic-rich soils on the cryogenically extracted
isotopic composition.
Future studies should test clay mineral fractionation effects on $\delta^{18}O$ and $\delta^2H$ during CWEs
individually. We further recommend running individual CWE spiking tests on each natural soil
material originating from field studies, also considering spatial variability of soil
physicochemical properties over depth. Comparing the isotopic deviation of results from such
spiking experiments with results from standardized soils will help to establish system-specific
transfer functions. This will require considerable effort. However, it seems to be the only way
to have some sort of calibration function for each extraction system and different soil types with
their clay mineral composition.
**5 Conclusions**
This work presents results from a worldwide round robin laboratory intercomparison test of
cryogenic extraction systems for soil water isotopic analysis. We tested the null hypothesis that
with identical soils, standards, and isotope analyses, cryogenic extraction across laboratories
should yield identical isotopic composition. The 16 participating laboratories used the same two
standard soils along with reference water of known isotopic composition for CWEs. With our
interlab comparison, we showed that multiple factors influence extracted isotopic signatures.
Soil type, water content, as well as the applied type of isotope analysis (OA-ICOS vs. IRMS),
showed major impacts, whereas, applied extraction parameters (time, temperature, and
vacuum) interestingly did not affect isotope results across laboratories. Laboratory internal
quality and water recovery rates showed additional effects.
Although the applied extraction system setups were different (e.g., size of extraction container,
heating unit), we could not show a major impact of the system's design on the isotope results,





as laboratories were successful for the one soil type and water content but failed for the other.
However, internal reproducibility of isotope results for the replicates of the same soil type at a
given WC was given for most of the labs. Nevertheless, different results were obtained for $\delta^{18}$O
and $\delta^2$H.
Our intercomparison work showed that defining any sort of standard extraction procedure for
CWEs across laboratories is challenging. Our results question the usefulness of this method as
a standard for water extraction since results are not intercomparable across laboratories. A
possible option might be that CWE labs establish system-specific calibration functions for each
natural soil type, individually, to correct for the given offset to a set of reference soils.
Finally, we note that while CWEs for soils are problematic for reasons discussed in this paper,
no work yet has seen any effects for plant water extractions apart from spectral interferences
when using OA-ICOS. However, such inter-laboratory and technique intercomparison should
be addressed for plants in the future to account for possible effects. New continuous, in-situ
measurements of soil and plant water isotopic composition might overcome isotope
fractionation issues we observed with CWE.



**Acknowledgements**
This interlaboratory test would not have been possible without the generous cooperation of the
researchers and technical staff in our 16 stable water isotope laboratories. We especially thank
Kim Janzen, Cody Millar, and Anna Winkler for their laboratory-support and Nathalie Steiner
for statistical support. The Gibson laboratory from Alberta Innovates Technology Futures is
thanked for IRMS analyses. This research was supported by an NSERC Discovery Grant and
Accelerator Award to J. J. McDonnell.





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




**Tables**
Table 1: Description of the respective extraction systems that participated in the cryogenic
inter-laboratory comparison and the applied extraction parameters for extraction approach I.
Note that not every lab provided the same detailed information.

| Lab no. | Country | Description of CWE facility | Number of extraction slots | Extraction parameters for approach I |
|---|---|---|---|---|
| 1 | Germany | Similar to lab No. 8; pair of Valco Exetainer® vials connected with a 1.56 mm stainless steel capillary as extraction-collection unit; a hot plate, LN$_2$-cold trap | 9 | Temperature: 100°C, vacuum: 1-6 Pa, time: 60 min (silty sand) 120 min (clayey loam) |
| 2 | Canada | Mainly composed of different types of Swagelok® fittings (Swagelok Company, Solon, OH, US), flanges, and flexible hoses (Rettberg®, Rettberg Inc., Göttingen, DE), vacuum applied or shut off via diaphragm valves and monitored via DCP 3000 and VSK 3000 (Vacuubrand Inc., Wertheim, DE), glass tubes as extraction and collection units, LN$_2$-cold trap, water bath/sand bath | 24 | Temperature: on average 96°C, vacuum: 3.3-7.3 Pa, time: 90 min (silty sand) 240 min (clayey loam) |
| 3 | Germany | Heating lamps; LN$_2$-cold trap | 5 | Temperature: ~115°C, vacuum: 1 Pa, time: 90 min |
| 4 | Germany | A septum-sealed 70 mL vial (extraction) and a Valco Exetainer® vial (collection) connected with a stainless steel capillary as extraction-collection unit; heating block (aluminum), LN$_2$-cold trap | 6 | Temperature: 125°C, vacuum: 50 Pa, time: 33 min (silty sand at 8% WC) and 56 min (silty sand at 20% WC), 67 min (clayey loam at 8% WC), 83 min (clayey loam at 20% WC) |
| 5 | France | Cold trap: mixture of LN$_2$ and EtOH | 4 | Temperature: 65°C, cold trap: -50--70°C, vacuum: 0.1-1 Pa (static vacuum), time: 60-90 min |
| 6 | Australia | Heating tape, glassware for extraction-collection unit; LN$_2$-cold trap | 4 | Temperature: 95-100°C, starting with sealed vacuum of 0.3 Pa, time: 150-180 min |
| 7 | Chile | Heating element: reactor HI 839800 (Hanna instruments); size of extraction container: 22mL; precision measured with VD81 Thyracont model | 9 | Temperature: 105°C, vacuum: 12-23 Pa, time: 240 min |
| 8 | Germany | Pair of Valco Exetainer® vials connected with a 1.56 mm stainless steel capillary as | 12 | Temperature: 200°C, vacuum: 50 Pa, time: 15 min |



| | | extraction-collection unit; an aluminum block on a hot plate, LN$_2$-cold trap | | |
|---|---|---|---|---|
| 9 | Germany | Stainless steel manifold (5 vials each), glass tubes as extraction-collection unit: 18 mm w, 150 mm l, LN$_2$-cold trap, water bath | 20 | Temperature: 95°C, vacuum: 0.8 Pa , time: 90 min |
| 10 | Switzerland | Glass tubes (Vacutainer), LN$_2$-cold trap, water bath | 20 | Temperature: 80°C |
| 11 | USA | Pyrex Culture Tubes (25mm x 150 mm), volume: 75 ml; heaters: electric coil (only allow to heat ⅔ of the tube) | 10 | Temperature: 102°C, vacuum: <0.1-2.7 Pa, time: on average 81 min (silty sand), 134 min (clayey loam) |
| 12 | Germany | Glass tubes, LN$_2$-cold trap, water bath | 8 | Temperature: 80°C, vacuum: 600 Pa, time: 60 min |
| 13 | Germany | Glass tubes (Schott GL 18), LN$_2$-cold trap, sensor-regulated tube-shaped heating element | 10 | Temperature: 100°C, vacuum: 6.7-13.3 Pa, time: 15-266 min |
| 14 | Germany | Glass tubes as extraction units, vacuum is generated by a Rotary vane pump (RZ 2.5, Vacuubrand, Wertheim, ) and monitored via DCP 3000 with VSP 3000 (Vacuubrand), LN$_2$-cold trap, water bath | 20 | Temperature: 80°C, vacuum: 2-46 Pa, time: 30 min (silty sand), 40 min (clayey loam) |
| 15 | Germany | The septa of Labco exetainers® are pierced with a cannula (1.2 mm diameter) and connected to the vacuum system, vacuum is generated by a Rotary vane pump (RZ 2.5, Vacuubrand, Wertheim, Germany) and monitored via DCP 3000 with VSP 3000 (Vacuubrand), LN$_2$-cold trap, water bath | 20 | Temperature: 80°C, vacuum: 10-350 Pa, time: 30 min (silty sand), 40 min (clayey loam) |
| 16 | Germany | Mainly composed of different types of Swagelok® fittings (Swagelok Company, Solon, OH, US), flanges, and flexible hoses (Rettberg®, Rettberg Inc., Göttingen, DE), vacuum applied or shut off via diaphragm valves and monitored via DCP 3000 and VSK 3000 (Vacuubrand Inc., Wertheim, DE), glass tubes as extraction and collection units, LN$_2$-cold trap, water bath/sand bath, high-purity nitrogen purging system | 18 | Temperature: 100°C, vacuum: 3.1-0.9 Pa, time: 45 min (silty sand), 240 min (clayey loam) |



Table 2: Soil characteristics of clayey loam and silty sand (means ± SD). The clay mineral composition
of soil samples was determined via X-ray powder diffraction (XRD, Philips X'Pert PW 1830 equipped
with a PW2273/20 tube and a theta/theta-goniometer) following Poppe et al. (2016). Values were not
corrected for reference intensity ratios (RIR). Alternating strata can occur for
Illite/Smectite/Vermiculite. X-ray fluorescence (XRF) characterization of the chemical composition (in
weight-%) was performed using an Axios spectrometer (PANalytical, EA Almelo, NL). Loss of ignition
was 12.8 for the clayey loam and 1.3 for the silty sand.

| Parameter | Clayey loam | Silty sand |
|---|---|---|
| pH-value | 7.2 ± 0.2 | 5.0 ± 0.3 |
| Water holding capacity [g 100g$^{-1}$] | 43.4 ± 0.8 | 32.1 ± 1.4 |
| Organic carbon [%] | 2.0 ± 0.2 | 0.7 ± 0.1 |
| Cation exchange capacity [cmol(+) kg$^{-1}$] | 30.6 ± 5.1 | 4.1 ± 0.6 |
| **Particle size [mm] distribution according to German DIN [%]** | | |
| <0.002 (clay) | 26 | 2.6 |
| 0.002–0.063 (silt) | 46.4 | 12.7 |
| 0.063–2 (sand) | 27.6 | 84.7 |
| **XRD analysis [relative %]** | | |
| Kaolinite | 18.8 | 18.8 |
| Illite | 18 | 27.7 |
| Chlorit | 1.2 | 19.8 |
| Vermiculite | 43.4 | 2.9 |
| Smectite | 0.5 | 19.8 |
| Mixed layered clays/alternating strata (Illite/Smectite/Vermiculite) | 18.1 | 11.1 |
| **XRF analysis [%]** | | |
| $SiO_2$ | 65.1 | 92.3 |
| $TiO_2$ | 0.4 | 0.1 |
| $Al_2O_3$ | 8.8 | 3.3 |
| $Fe_2O_3$ | 3.1 | 0.5 |
| MnO | 0.1 | 0.0 |
| MgO | 1.5 | 0.1 |
| CaO | 5.3 | 0.2 |
| $Na_2O$ | 0.9 | 0.3 |
| $K_2O$ | 1.7 | 1.7 |
| $P_2O_5$ | 0.2 | 0.1 |
| $SO_3$ | 0.1 | <0.01 |
| Cl | <0.002 | <0.002 |
| F | <0.05 | <0.05 |





**Figures**
Figure 1. Water recovery rates (grouped from <80 to >98%) for both soil types (clayey loam
and silty sand), WCs (8% and 20%) and OA-ICOS and IRMS (upper and lower panels,
respectively) isotope results in comparison to the spiked reference DI water (red asterisks)
shown in dual isotope space. For reference, plots include the Global Meteoric Water Line
(GMWL, solid red line). Water recovery rates are shown for those labs that provided the
complete set of soil weight data (in % of previously added water).

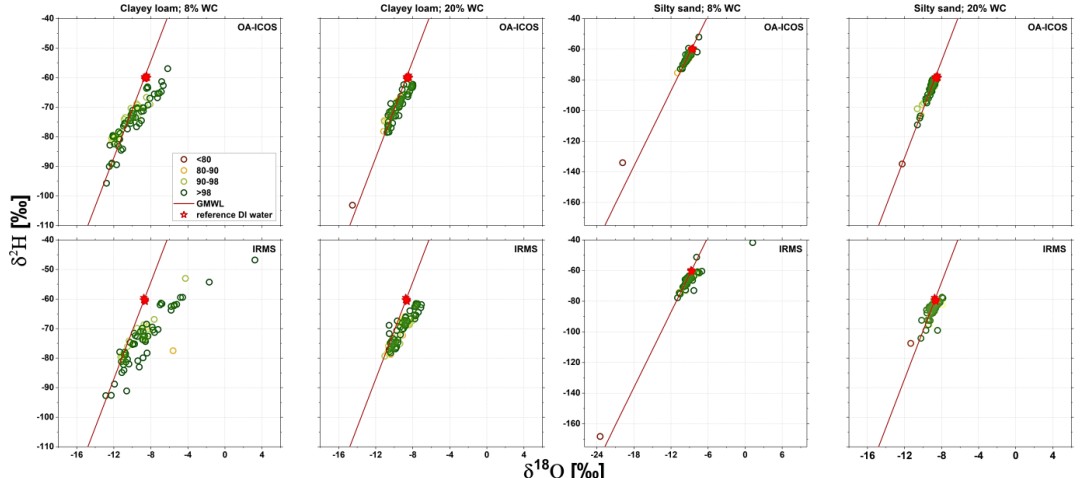





1    Figure 2. Effect of cryogenic extraction parameters (duration, temperature, and pressure) on

2    $\delta^2$H results of both soil types (clayey loam and silty sand) and WCs (8 and 20%) shown for all

3    labs. The mean reference DI water $\delta^2$H value is included as red dotted line.

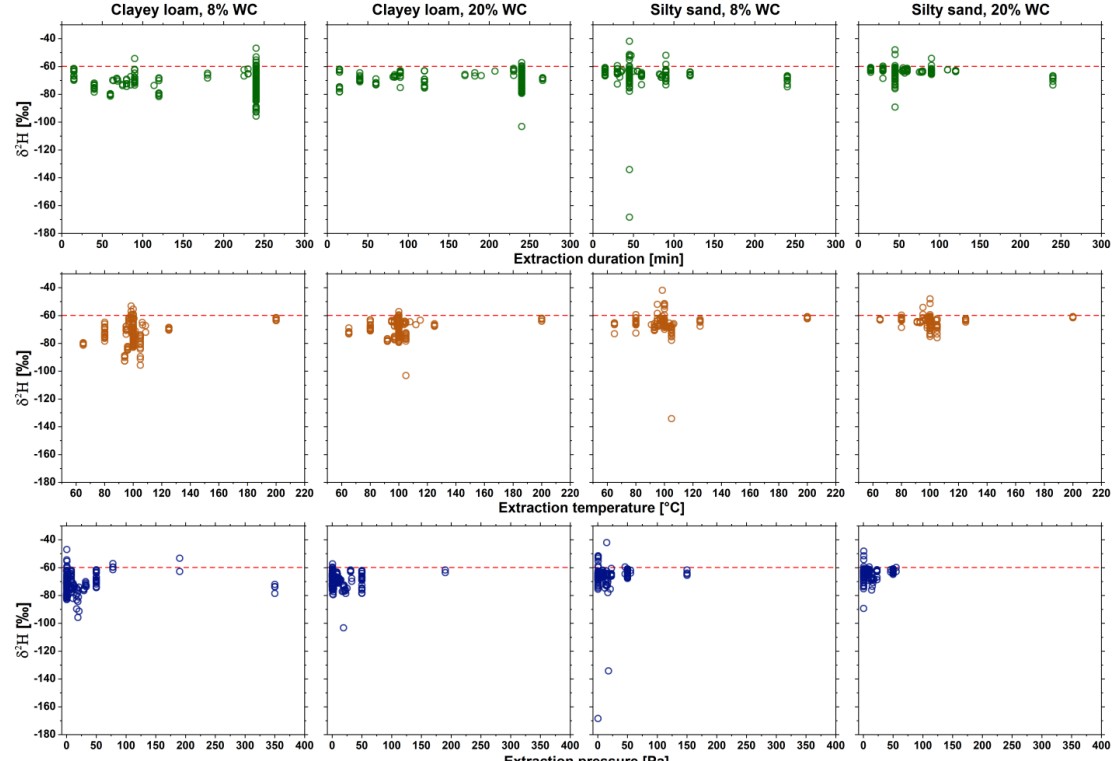





Figure 3. Mean differences from reference DI water for $\delta^2$H OA-ICOS results of water extracts
from both extraction methods (lab-procedure: I and pre-defined: II), soil types, and water
contents (8 and 20% WC) including TSD of ±2 for $\delta^2$H (Asterisk: -108.4 for $\delta^2$H). Y-error bars
represent the isotopic variation of the replicates. There were no significant differences between
the two extraction approaches overall labs.

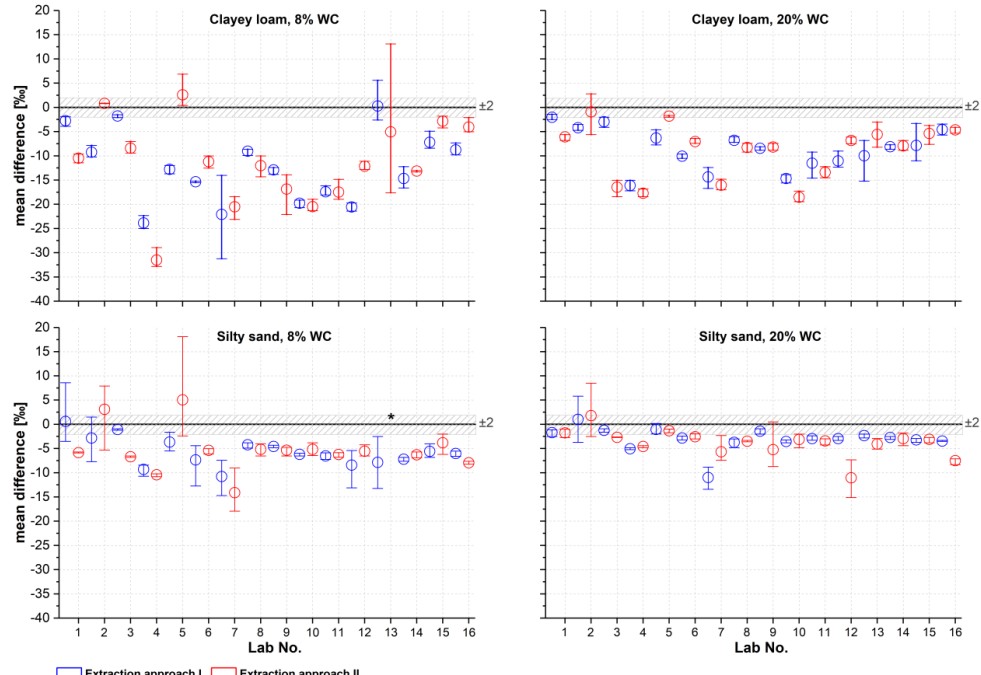





Figure 4. Mean differences from reference DI water for $\delta^{18}O$ OA-ICOS results of water extracts
from both extraction methods (lab-procedure: I and pre-defined: II), soil types, and water
contents (8 and 20% WC) including TSD of ±0.2 for $\delta^{18}O$. Asterisks represent outliers. Y-error
bars represent the isotopic variation of the replicates. There were no significant differences
between the two extraction approaches overall labs.

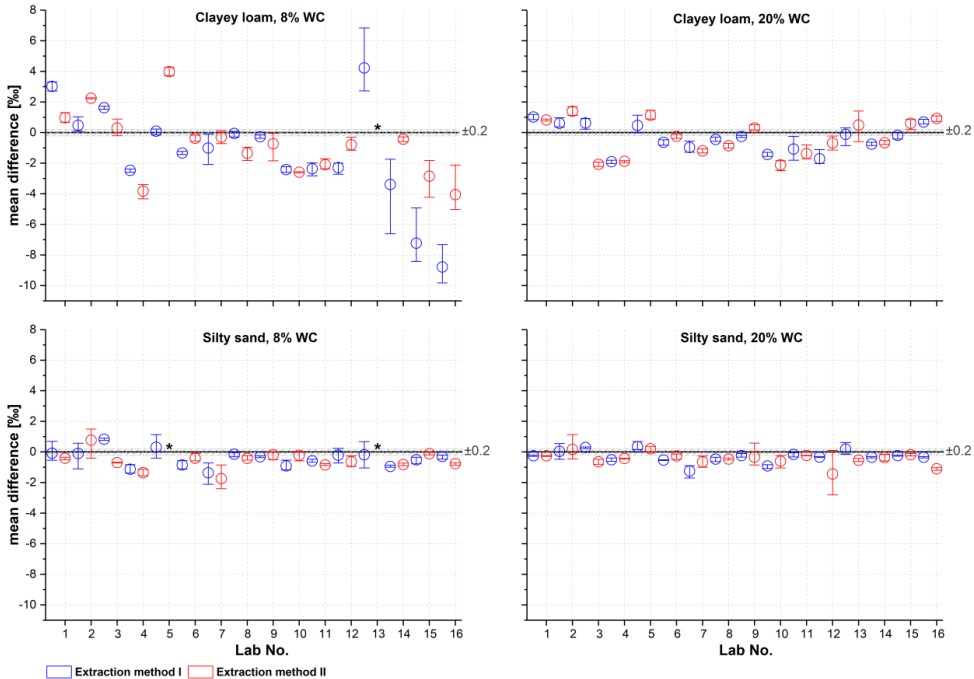





Figure 5. Dual isotope plots of clayey loam extracts for 8% and 20% WC in comparison to
reference DI water (red asterisks) for OA-ICOS and IRMS (upper and lower panels,
respectively). For reference, plots include the Global Meteoric Water Line (GMWL, solid red
line) and evaporation water lines for 8% and 20% WC (solid green and orange lines,
respectively).

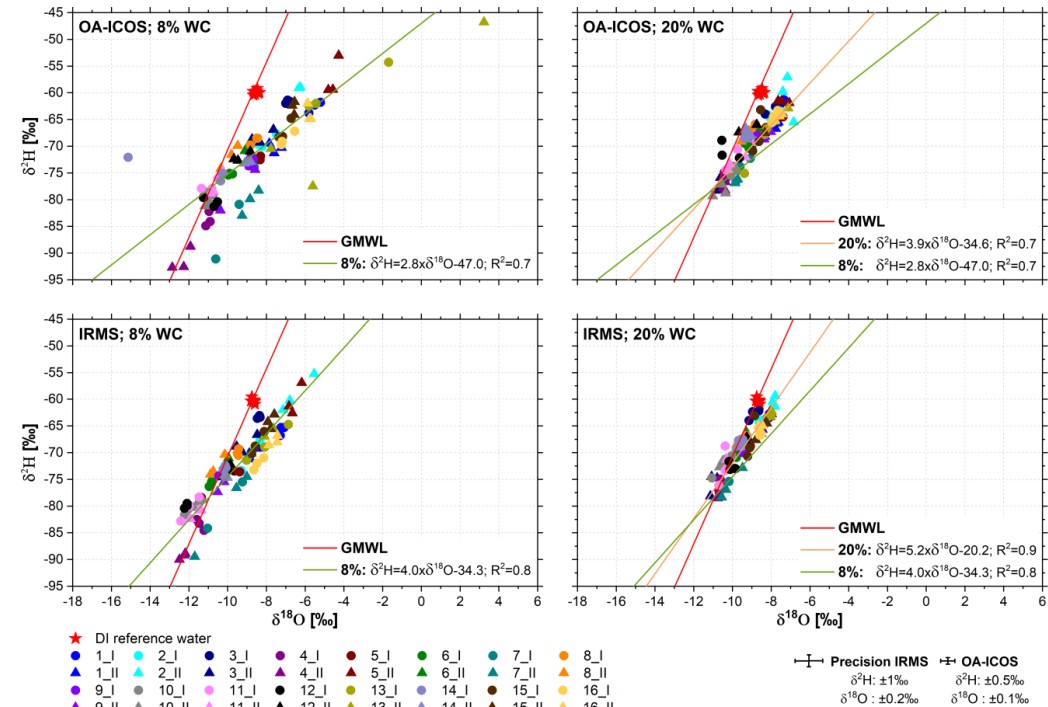





Figure 6. Dual isotope plots of silty sand extracts for 8% and 20% WC in comparison to
reference DI water (red asterisks) for OA-ICOS and IRMS (upper and lower panels,
respectively). For reference, plots include the Global Meteoric Water Line (GMWL, solid red
line) and evaporation water lines for 8% and 20% WC (solid green and orange lines,
respectively).

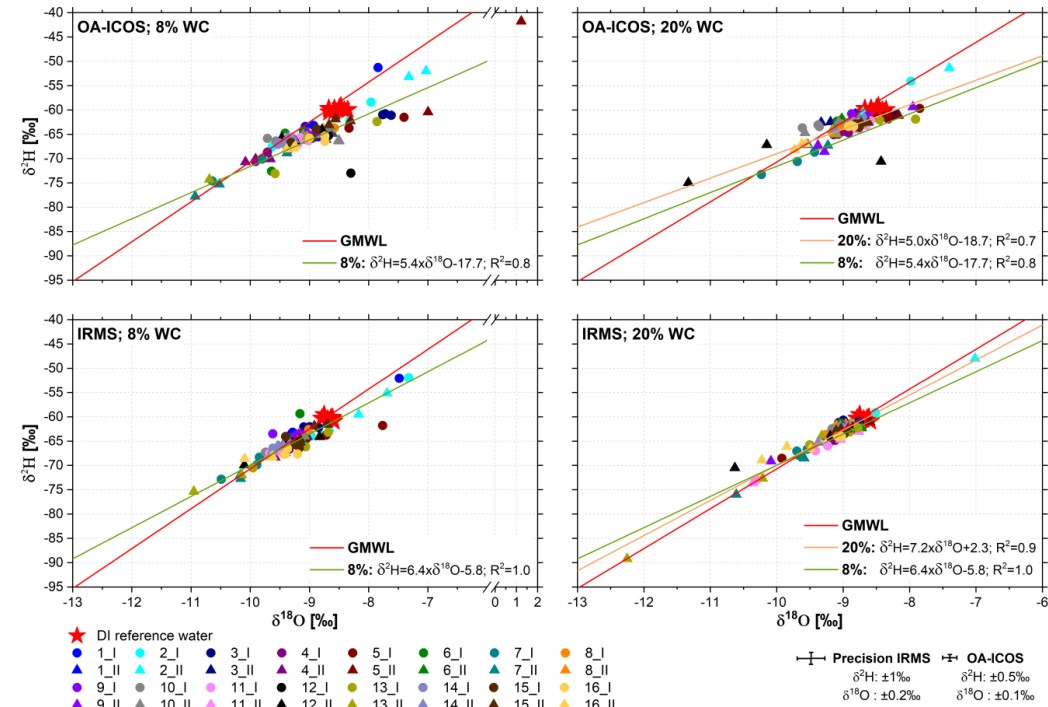





Figure 7. Dual isotope plots of clayey loam and silty sand extracts for 8% and 20% WC in
comparison to reference DI water (red asterisks) for OA-ICOS analyses flagged by spectral
contamination using the Spectral Contamination Identifier (LWIA-SCI) post-processing
software (Los Gatos Research Inc.). BB-NB: Broad-and narrow-band absorbers (ethanol and
methanol); NB: narrow-band absorber (methanol); NC: no contamination detected.

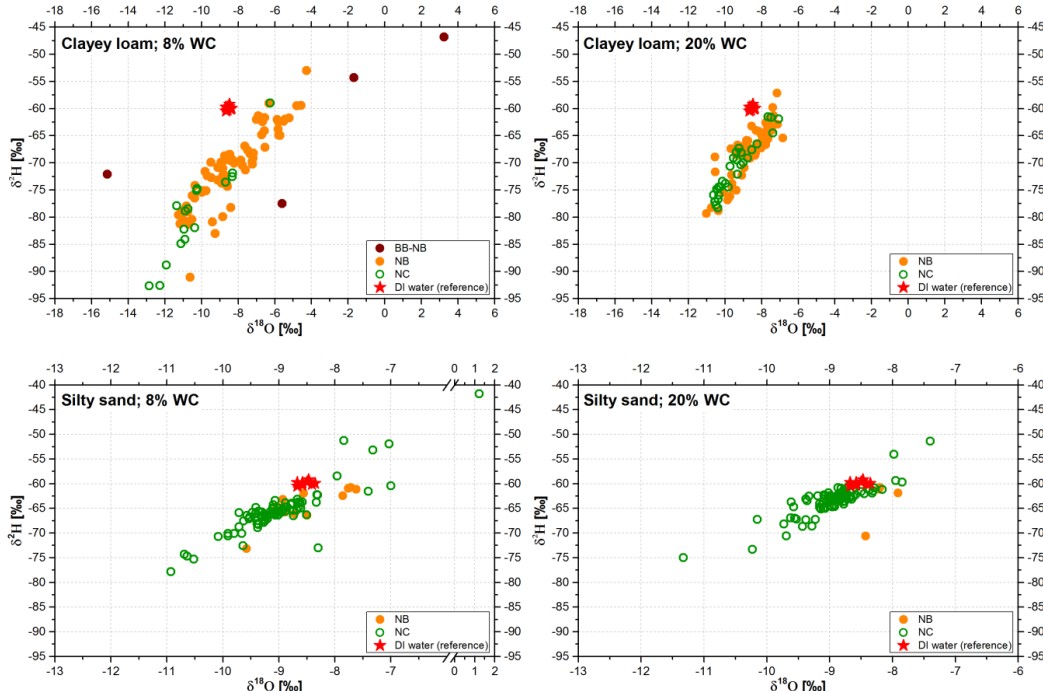



**Appendix**
Appendix 1. Cryogenic system – Questionnaire
**Inter-laboratory comparison of CWE systems**

Contact person

Last Name          First Name

Address    Street          Street No.

City          Postal Code    Country

Phone Number       Email

**Cryogenic system – Questionnaire**
How many numbers of extraction slots/units does your cryogenic extraction system have?
How much sample material (in gramm) is required for the cryogenic extraction at your system?
Does your laboratory have an operating procedure in terms of temperature, vacuum settings, and
extraction times for soil and plant samples?
Do you have the possibility to adjust the extraction conditions (temperature, vacuum)?
To which type of sample material do you apply the cryogenic extraction method?

Type of plant material (e.g., twig, root crown)     Soil type

Please provide us a photo of your cryo-line.

