# Peer review of "Inter-laboratory comparison of cryogenic water extraction systems for stable isotope analysis of soil water"

_Hydrology and Earth System Sciences, 2018_

## Referee Comment (RC1) · N. Munksgaard (Referee) · 26 Mar 2018

This inter-laboratory comparison study presents very significant findings regarding the performance of cryogenic water extraction (CWE) systems for soil water stable isotope analysis. The paper is well written with clear findings and illustrations. The lead and senior authors have developed and researched CWE system for several years and have previously published several papers on these systems (e.g. Orlowski et al. 2013, 2015, 2016). The present study represents a welcome initiative to help improve analytical techniques. The comparison study was well designed with the emphasis on documenting the difference between the known isotopic composition of a supplied wa-

ter sample and the extracted isotopic composition of soil water derived from the known water by using it to wet dry soils. The key finding is the surprisingly large difference in performance of CWE systems in different laboratories (Fig. 3 and 4). A few laboratories performed relatively well but none were acceptable (as per the study criteria) for all soils, water contents and isotope systems. The fact that most laboratories performed very poorly for one or both soils using a system that has been regarded as the main stay of soil (and plant) water isotope analysis is a disturbing conclusion (the authors note their dismay). → What are the consequences for the reliability of the numerous previous studies relying on these and similar laboratories? Were studies conducted without the rigorous quality control (recovery of known water isotopic composition) carried out in the present study? These questions should be noted (if not answered) in the discussion. The interlaboratory comparison confirmed the influence of many factors that affect accuracy as documented in previous publications. However, the lack of systematic relationships between isotopic recovery of soil waters and CWE parameters prevented clear conclusions from being drawn regarding which future steps can be taken to improve performance. This suggest that a complex interaction of many factors including soil type, temperature, vacuum etc. influence CWE results. These may also include the specific design and operation of each CWE system. These findings are also highly significant in light of the Orlowski et al 2016 study (Hydrological Processes: Intercomparison of soil pore water extraction methods for stable isotope analysis. Natalie Orlowski, Dyan L. Pratt and Jeffrey J. McDonnell) which compared five different techniques for analysing soil water isotope composition. It appears that the differences in accuracy of these five techniques were no larger than the difference in accuracy between the sixteen CWE systems presented in the present study. → This raises the possibility that the success of any of these techniques may depend more on the specific understanding, design and settings/operation of each technique than an inherent superiority of one technique over another. This aspect should be added to the discussion. Specific comments: P5 L20: I understand that choices had to be made but it should be acknowledged that the drying and rewetting steps may have influenced the

outcomes if not performed the same way - were instructions on these steps included? P5 L27-32: I can't locate the data from this reliability test, were all labs successful in this test? P6 L14: "alternating fashion" - it is unclear exactly how this step was carried out P6 L14: How much soil was loaded by each lab (both using their own method and the prescribed method)? The questionnaire asked this question, but the information doesn't seem to be presented. Is it possible that soil inhomogeneity was a factor if a lab used small amounts? P13 L27: Incomplete drying before wetting may also have led to >100% recovery during CWE P14 L5: Freezing of the wetted soils before loading in the CWE may reduce vapour loss during evaporation P17 L22-23: This sentence seems adrift here, but it is a valid point that should be expanded upon, possibly in the introduction/background. It is a valid question to ask whether it is actually relevant to extract all water from a soil sample - it will depend on the study context. P17 L29: Not all laser instruments (LAS) were of the OA-ICOS type (Los Gatos) in the WICO study (Wassenaar et al 2018) - several were CRDS instruments (Picarro). This section should also be modified with respect to organic interferences in LAS, the effects can be dramatically different between Los Gatos and Picarro instruments both in direction and magnitude. They are even different between different generations of Picarros (see e.g. Munksgaard et al. Rapid Commun. Mass Spectrom. 2014, 28, 2151–2161) P18 L28: Interferences can also be overcome by in-line high temperature oxidation prior to LAS measurement, although this will likely contribute small amounts of H2O which may or may not be significant compared to the overall extraction amount. P20 L11: Does this mean that in effect each soil would have to be investigated (i.e. a standard addition technique) unless a series of samples have very similar contents and type of organics and clay? - a very tedious process. P21 L13: Can the authors expand on what these techniques could look like?

---

## Referee Comment (RC2) · Anonymous Referee #2 · 11 Apr 2018

This work provides a good initiative towards the standardization of a procedure that can be carried out in multiple ways. The inclusion of laboratories worldwide depicts this need as well as the large variety of extraction systems developed to carry out the CWE. The idea of involving multiple laboratories under the same approach is well done and supported with a good protocol (despite the possible sources of errors described by the authors). However, even if the authors do not mention a specific method or guidelines at the end of the paper; the information provided can lead to the best practice.

On the Methods (page 5, lines: 27-32) and Discussion (page 14, lines: 15-22) sections, the authors mention the "performance test" carried out by the laboratories. The analysis

of this data needs more attention on the Results section and it can be integrated with a $Z_{score}$ graph as the one showed by Orlowski et al (2016). This type of plot will help to see the proportional laboratory efficacy to reach the labeled water. In addition, it will be important to add information about the performance test carried out by the laboratories, because the paper only mentions the data of two laboratories and leave on the dark the data from the other 14.

The laboratory capacity to successfully extract soil water is essential for most of the projects relying on that information (the reason why the authors defined the experiment). This work shows that despite having identical soil and water samples, as well as the protocol extraction ("pre-defined extraction method"), no one of the laboratories was able to reach the water signature. The proposed rehydration process could be affected by small differences among the soil samples sent to each laboratory (non-homogeneous composition between subsamples of the same soil). This brings the question if different subsamples of soil were analyzed to test the homogeneity among samples as the authors did with the water? In addition, did any of the laboratories send a sample of rehydrated soil for its physical and chemical analysis? Because this can help the authors to support their assumption (Discussion section, page 17; 13-15)

Despite the authors sentence (Discussion section, page 20, lines: 1-2): "We found no clear tendency for which approach should be applied, thus at present, and much to our dismay, we cannot define any standard protocol for CWE"; the information contained in this paper can give important clues about the feasibility of applying one specific method. If the authors apply the $Z_{score}$ graph (Orlowski et al, 2016) mentioned previously (second paragraph); they can determine which methods lead towards a more accurate extraction among all the setups evaluated considering the pre-defined protocol (analysis 1) and considering the laboratory protocol (analysis 2). In this way, the authors can provide as "take-home messages" the laboratory practices that lead towards better results.

If the authors change the notation from 20

The amount of sample material used per laboratory is not reflected in table 1 and this information can help to understand the differences.

References

Orlowski, Natalie, Dyan L. Pratt, and Jeffrey J. McDonnell. "Intercomparison of soil pore water extraction methods for stable isotope analysis." Hydrological Processes 30.19 (2016): 3434-3449.
* * *

---

## Referee Comment (RC3) · D. Penna (Referee) · 20 Apr 2018

General comment:

This manuscript reports the results of an intercomparison exercise that aimed at testing the consistency of cryogenic water extractions for the analysis of stable isotope of hydrogen and oxygen in soil water among worldwide-distributed laboratories. In the last few years, the ecohydrological and soil science international communities have shown a strong and increasing interest in better understanding the functional interrelationships between soil and vegetation based on the use of stable water isotope data. The cryogenic water extraction technique has so become a golden standard for sampling water

from the unsaturated zone. Very recent studies, often conducted by the first author of this manuscript and colleagues, already showed some potentials and limitations of this technique, and provided helpful information for users. However, a worldwide interlaboratory comparison among several cryogenic extraction facilities was still missing, thus this work it is certainly welcome. Indeed, I believe that this manuscript is timely and of great interest for the readers of this journal.

The manuscript is very well written, logically structured, nicely illustrated, and the conceptual steps can be followed very well. The working hypothesis and the specific objectives are well posed, following a substantial introduction, the statistical analysis is correct, and the results and interpretation are well supported by the data. The methodological approach leading to the comparison exercise was solidly defined and clearly presented.

As noted by the Authors themselves, the large differences in performance found among the labs included in the exercise are somehow worrisome and pose questions on the possible adoption of cryogenic water extraction as a standard method for soil water sampling. However, these results are very relevant to the scientific community because implicitly suggest cautions in comparing isotope soil water data extracted by different facilities, and indicate that much technical work is still needed to test possible further controls on these differences and develop new techniques able to return more consistent results.

I have only the following minor comments for the Authors to address.

Minor comments and technical corrections:

P4 L11. Here, and throughout the rest of the manuscript (e.g., P7 L16; P9 L12-14; P11 L15…), it is not immediately clear what "isotope results" are, and I suggest to replace this term with "values" or "data".

P5 L14. The authors reported that the soil was homogenized before shipping. However, as noted by the other reviewers, I wonder if possible heterogeneities in the analysed soil samples (especially for small volumes) could have been present and could have affected the results.

P5 L27. I find the definition of "water-water" cryogenic extraction a bit confusing. I suggest to use, throughout the manuscript, simply the terms "water extraction" vs. "soil water extraction", or something similar.

P7 L1-2. Although intuitive, I suggest to add a short explanation about the choice of applying different extraction times for the silty sand and the clayey loam soils.

P8 L28, and P12 L6-20, and Figs. 5-6. It seems to me that these are not "true evaporation lines" describing the progressive isotopic enrichment of an individual source water (see Benettin et al., 2018 who reported the often misused term and concept of evaporation line). This might not be a critical point in the interpretation of the results and the overall meaning of the research. However, for the sake of accuracy, I suggest to check this and in case change the terminology (eg, simply calling them regression lines) and slightly re-interpret the results reported at P12 L6-20. Moreover, it's not very clear to me why in the left panels of Figs. 5 and 6 (8% WC) one regression line (for 8% samples) is reported in addition to the GMWL whereas in right panels of Figs. 5 and 6 (20% WC) two regression lines are shown (both for 8% and 20% WC). Please, fix this or explain.

Benettin, P., et. al. Effects of climatic seasonality on the isotopic composition of evaporating soil waters, Hydrol. Earth Syst. Sci. Discuss., https://doi.org/10.5194/hess-2018-40, in review, 2018.

P9 L23-24. Do the Authors have any idea about the reason for recovery rates higher than 100%? Could this somehow affect the results? Perhaps a sentence could be added here (trying to avoid the risk of speculation).

P11 L27. In addition to the statistical results, I wonder whether it might be appropri-

ate to show OA-ICOS and IRMS data as boxplots to graphically stress the difference between values returned by the two techniques.

P17 L23. The reference of "Orlowsky et al., 2018" is missing from the reference list.

Fig. 1. I suggest to increase the size of the axis labels.

Fig. 3 and Fig. 4. In the caption: was the mean computed among the three replicates? If so, I suggest to specify this for the sake of clarity.

Fig. 5 and Fig. 6. I suggest to add in the caption that the legend includes explanation for the symbols used for the 16 labs and the two extraction approaches.

---

## Author Comment (AC1) · 30 May 2018

**Reply to Reviewer 1**

In the following, please find the corrections and comments to the referee's response.

**General comments**

This inter-laboratory comparison study presents very significant findings regarding the performance of cryogenic water extraction (CWE) systems for soil water stable isotope analysis. The paper is well written with clear findings and illustrations. The lead and senior authors have developed and researched CWE system for several years and have previously published several papers on these systems (e.g. Orlowski et al. 2013, 2015, 2016). The present study represents a welcome initiative to help improve analytical techniques. The comparison study was well designed with the emphasis on documenting the difference between the known isotopic composition of a supplied water sample and the extracted isotopic composition of soil water derived from the known water by using it to wet dry soils. The key finding is the surprisingly large difference in performance of CWE systems in different laboratories (Fig. 3 and 4). A few laboratories performed relatively well but none were acceptable (as per the study criteria) for all soils, water contents and isotope systems. The fact that most laboratories performed very poorly for one or both soils using a system that has been regarded as the main stay of soil (and plant) water isotope analysis is a disturbing conclusion (the authors note their dismay). ! What are the consequences for the reliability of the numerous previous studies relying on these and similar laboratories? Were studies conducted without the rigorous quality control (recovery of known water isotopic composition) carried out in the present study? These questions should be noted (if not answered) in the discussion. The interlaboratory comparison confirmed the influence of many factors that affect accuracy as documented in previous publications. However, the lack of systematic relationships between isotopic recovery of soil waters and CWE parameters prevented clear conclusions from being drawn regarding which future steps can be taken to improve performance. This suggest that a complex interaction of many factors including soil type, temperature, vacuum etc. influence CWE results. These may also include the specific design and operation of each CWE system. These findings are also highly significant in light of the Orlowski et al 2016 study (Hydrological Processes: Intercomparison of soil pore water extraction methods for stable isotope analysis). Natalie Orlowski, Dyan L. Pratt and Jeffrey J. McDonnell which compared five different techniques for analysing soil water isotope composition. It appears that the differences in accuracy of these five techniques were no larger than the difference in accuracy between the sixteen CWE systems presented in the present study. ! This raises the possibility that the success of any of these techniques may depend more on the specific understanding, design and settings/operation of each technique than an inherent superiority of one technique over another. This aspect should be added to the discussion.

**General comments**

**Response:** We thank Niels Munksgaard for taking the time to review our manuscript and providing this generally positive feedback.

Most past studies that applied CWE did not or barely provide any information on the applied extraction parameters and did not carry out any sort of quality control of the system's reliability (as per Orlowski et al. (2013)) and the obtained CWE isotope data (as per Gaj et al. (2017)). Therefore, possible fractionation effects associated with CWE remain unknown for most past studies. As we already argued in our recent paper (Orlowski et al., 2018), when CWE data is used to calculate plant's

water source, errors could be quite large and lead to misinterpretations of the role different plant species play in hydrologic processes at the ecosystem or larger scales (Zhao et al., 2016). However, in order to understand from which soil water source plants take up their water, we need to have a sound understanding of the interactions between water (mobile and higher tension water) and the overall soil compartment. Current lab-based water extraction techniques (not only CWE) remain one of the biggest challenges in achieving this goal (Orlowski et al., 2018). We will add this argumentation to the revised manuscript version.

We will further add the following to the discussion section of the revised manuscript: "In the light of our experience with other soil water extraction techniques (Orlowski et al., 2016b), we argue that the success of any of these methods may depend more on the specific understanding and operation leading to internal reproducibility of each individual technique's results than an inherent superiority of one technique over another."

**Specific comments**

1. P5 L20: I understand that choices had to be made but it should be acknowledged that the drying and rewetting steps may have influenced the outcomes if not performed the same way - were instructions on these steps included?

**Response:** We already discussed this point in our previous version of the manuscript: "Again, it should be stressed here that for our intercomparison soil samples were oven-dried twice (before and after shipment) prior to any rewetting and labs were advised to store the dried samples in a desiccation chamber until use…sample preparation might have played its role, when it comes to discrepancies in lab's results… Remoistening of oven-dried soil samples might be a general problem of such spiking experiments….However, so far, regular oven-drying of soils is standard practice (Koeniger et al., 2011) for such rewetting experiments in the literature."

2. P5 L27-32: I can't locate the data from this reliability test, were all labs successful in this test?

**Response:** We did not include these results in our manuscript since not every lab provided the full set of data. We therefore picked some examples.

3. P6 L14: "alternating fashion" - it is unclear exactly how this step was carried out

**Response:** We will revise the sentence as follows: "Soil and DI water were added alternately."

4. P6 L14: How much soil was loaded by each lab (both using their own method and the prescribed method)? The questionnaire asked this question, but the information doesn't seem to be presented. Is it possible that soil inhomogeneity was a factor if a lab used small amounts?

**Response:** We will add this information to Table 1 and the respective results section: "In relation to the amount of used sample material, most labs either introduced 10 or 20 g to their system no matter the extraction approach (I or II), soil type or WC. Only labs 11 and 13 chose different weights with respect to the WC, e.g., 10 g for the higher WC (20%) and 20 g for 8% WC for extraction approach I."

5. P13 L27: Incomplete drying before wetting may also have led to >100% recovery during CWE

**Response:** We included this point in the previous manuscript version: "Again, it should be stressed here that for our intercomparison soil samples were oven-dried twice (before and after shipment) prior to any rewetting and labs were advised to store the dried samples in a desiccation chamber until use." And further: "However, oven-drying was performed at an intermediate temperature (105°C for 48h) and not under vacuum as per Savin and Epstein (1970) and different indoor laboratory 'climatic conditions' at the participating laboratories were observed. Thus, it might be possible that not all of the clay interlayer and adsorbed water was removed or made isotopically non-exchangeable, and that non-equilibrium isotopic fractionation occurring at different temperatures during heating might be responsible for some of the differences we observed."

6. P14 L5: Freezing of the wetted soils before loading in the CWE may reduce vapour loss during evaporation

**Response:** We did not include any recommendations on freezing the samples before water extraction. However, we agree with the reviewer and most labs indicated that they have frozen the samples before the actual water extraction to prevent evaporative water losses.

7. P17 L22-23: This sentence seems adrift here, but it is a valid point that should be expanded upon, possibly in the introduction/background. It is a valid question to ask whether it is actually relevant to extract all water from a soil sample - it will depend on the study context.

**Response:** We will discuss this aspect in more detail and shift it to the discussion section.

8. P17 L29: Not all laser instruments (LAS) were of the OA-ICOS type (Los Gatos) in the WICO study (Wassenaar et al 2018) - several were CRDS instruments (Picarro). This section should also be modified with respect to organic interferences in LAS, the effects can be dramatically different between Los Gatos and Picarro instruments both in direction and magnitude. They are even different between different generations of Picarros (see e.g. Munksgaard et al. Rapid Commun. Mass Spectrom. 2014, 28, 2151–2161)

**Response:** We will edit this paragraph following the reviewer's suggestions.

9. P18 L28: Interferences can also be overcome by in-line high temperature oxidation prior to LAS measurement, although this will likely contribute small amounts of $H_2O$ which may or may not be significant compared to the overall extraction amount.

**Response:** We will add this point to the discussion section: "Martín-Gómez et al. (2015) introduced an on-line oxidation method for organic compounds for samples measured via isotope-ratio infrared spectroscopy. The authors showed that this method was able to effectively remove methanol interference, but was not efficient for high concentrations of ethanol."

10. P20 L11: Does this mean that in effect each soil would have to be investigated (i.e. a standard addition technique) unless a series of samples have very similar contents and type of organics and clay? - a very tedious process.

**Response:** So far, we do not see any other possibility.

**References**

Gaj, M., Kaufhold, S., Koeniger, P., Beyer, M., Weiler, M., & Himmelsbach, T. (2017). Mineral mediated isotope fractionation of soil water. *Rapid Communications in Mass Spectrometry*, *31*(3), 269–280. https://doi.org/10.1002/rcm.7787

Koeniger, P., Marshall, J. D., Link, T., & Mulch, A. (2011). An inexpensive, fast, and reliable method for vacuum extraction of soil and plant water for stable isotope analyses by mass spectrometry. *Rapid Communications in Mass Spectrometry*, *25*(20), 3041–3048. https://doi.org/10.1002/rcm.5198

Martín-Gómez, P., Barbeta, A., Voltas, J., Peñuelas, J., Dennis, K., Palacio, S., et al. (2015). Isotope-ratio infrared spectroscopy: a reliable tool for the investigation of plant-water sources? *New Phytologist*, 1–14. https://doi.org/10.1111/nph.13376

Orlowski, N., Frede, H.-G., Brüggemann, N., & Breuer, L. (2013). Validation and application of a cryogenic vacuum extraction system for soil and plant water extraction for isotope analysis. *Journal of Sensors and Sensor Systems*, *2*(2), 179–193. https://doi.org/10.5194/jsss-2-179-2013

Orlowski, N., Winkler, A., McDonnell, J. J., & Breuer, L. (2018). A simple greenhouse experiment to explore the effect of cryogenic water extraction for tracing plant source water. *Ecohydrology*, e1967. https://doi.org/10.1002/eco.1967

Savin, S. M., & Epstein, S. (1970). The oxygen and hydrogen isotope geochemistry of clay minerals. *Geochimica et Cosmochimica Acta*, *34*(1), 25–42. https://doi.org/10.1016/0016-7037(70)90149-3

Zhao, L., Wang, L., Cernusak, L. A., Liu, X., Xiao, H., Zhou, M., & Zhang, S. (2016). Significant Difference in Hydrogen Isotope Composition Between Xylem and Tissue Water in Populus Euphratica. *Plant, Cell & Environment*, *39*(8), 1848–1857. https://doi.org/10.1111/pce.12753

---

## Author Comment (AC2) · 30 May 2018

**Reply to Reviewer 2**

In the following, please find the corrections and comments to the referee's response.

This work provides a good initiative towards the standardization of a procedure that can be carried out in multiple ways. The inclusion of laboratories worldwide depicts this need as well as the large variety of extraction systems developed to carry out the CWE. The idea of involving multiple laboratories under the same approach is well done and supported with a good protocol (despite the possible sources of errors described by the authors). However, even if the authors do not mention a specific method or guidelines at the end of the paper; the information provided can lead to the best practice. On the Methods (page 5, lines: 27-32) and Discussion (page 14, lines: 15-22) sections, the authors mention the "performance test" carried out by the laboratories. The analysis of this data needs more attention on the Results section and it can be integrated with a Zscore graph as the one showed by Orlowski et al (2016). This type of plot will help to see the proportional laboratory efficacy to reach the labeled water. In addition, it will be important to add information about the performance test carried out by the laboratories, because the paper only mentions the data of two laboratories and leave on the dark the data from the other 14. The laboratory capacity to successfully extract soil water is essential for most of the projects relying on that information (the reason why the authors defined the experiment). This work shows that despite having identical soil and water samples, as well as the protocol extraction ("pre-defined extraction method"), no one of the laboratories was able to reach the water signature. The proposed rehydration process could be affected by small differences among the soil samples sent to each laboratory (nonhomogeneous composition between subsamples of the same soil). This brings the question if different subsamples of soil were analyzed to test the homogeneity among samples as the authors did with the water? In addition, did any of the laboratories send a sample of rehydrated soil for its physical and chemical analysis? Because this can help the authors to support their assumption (Discussion section, page 17; 13-15) Despite the authors sentence (Discussion section, page 20, lines: 1-2): "We found no clear tendency for which approach should be applied, thus at present, and much to our dismay, we cannot define any standard protocol for CWE"; the information contained in this paper can give important clues about the feasibility of applying one specific method. If the authors apply the Zscore graph (Orlowski et al, 2016) mentioned previously (second paragraph); they can determine which methods lead towards a more accurate extraction among all the setups evaluated considering the pre-defined protocol (analysis 1) and considering the laboratory protocol (analysis 2). In this way, the authors can provide as "take-home messages" the laboratory practices that lead towards better results. If the authors change the notation from 20. The amount of sample material used per laboratory is not reflected in table 1 and this information can help to understand the differences.

**Response:** We thank referee #2 for the very valuable and positive comments, which helped us to improve our manuscript.

Figures 3 and 4 provide mean differences to the reference DI water allowing assumptions on how good or bad individual labs performed with respect to the "reference". Simply put, a Z-score is a numerical measure of a value's relationship to the mean in a group of values. If a Z-score is 0, it represents the score as identical to the mean score. Figure 3 and 4 provide this information, not in dual isotope space but individually for each isotope. The figures further include the SD of the lab's

extraction results. It is this information that the paper intends to convey. We therefore would prefer, and we hope our argument above helps to demonstrate this, not to provide additional Z-score plots for the soil water extraction results. They would be redundant in some ways and would not convey the precise message we intend with Figures 3 and 4 in their current form.

With respect to the "performance test" results, we did not include these results in our manuscript since not every lab provided the full set of data. We therefore picked some examples as the data allowed. Again, a great suggestion but our data limited us in this regard.

We can rule out any inhomogeneity of the bulk soil sample's soil physico-chemical properties. The LUFA Speyer provides "standard soils" exactly for conducting lab studies (https://www.lufa-speyer.de/index.php/dienstleistungen/standardboeden/8-dienstleistungen/artikel/57-standard-soils). The LUFA already dried, homogenized and sieved (with a 2 mm screen) the bulk soil sample according to "Good Laboratory Practice". However, we repeated these steps before taking subsamples for the individual labs. Since every lab rehydrated the soil samples with the same deionized water (that we sent to them) and soils were only used once per extraction, we assume that potential changes in the soil properties due to the water additions are the same across labs. We did not see the necessity in determining the soil properties for a second time after water extraction, since samples were not extracted multiple times. Soil properties were determined according to German DIN-ISO norms on dried soil samples (e.g., DIN-ISO 11465, DIN-ISO 10390). We therefore did not see the need in analysing soil properties on a rehydrated soil sample.

With respect to the "take-home-message", we will further add the following: "Orlowski et al. (2018) recently explored the effect of CWE for tracing plant source water. The authors tested the ability to match plant water to its putative soil water source(s) by using different CWE conditions (30–240 min, 80–200 °C, 0.1 Pa) for a clayey loam (same as in this study) and a sand. They showed that with higher extraction temperatures and longer extraction times, gradually more enriched soil water was extracted, which surprisingly reflected the plants' source water…For most past studies, possible fractionation effects associated with CWE remain unknown and the applied extraction parameters or cryogenic system specifications are often not indicated. Orlowski et al. (2018) recently stated that observed isotopic fractionation effects potentially lead to errors when CWE isotope data is used for plant water source calculation. This miscalculation in plant's water source could be quite large and could lead to misinterpretations of the role different plant species play in hydrologic processes at the ecosystem or larger scales. Millar et al. (2018) used the most common water extraction methods (centrifugation, microwave extraction, direct vapor equilibration, high-pressure mechanical squeezing, and two different CWE systems) for their intercomparison study on spring wheat (Triticum aestivum L.). The authors showed that all methods yielded markedly different isotopic signatures. The various methods also produced differing concentrations of co-extracted organic compounds. Again, CWE was outperformed by other extraction methods...In the light of our experience with other soil water extraction techniques (Orlowski et al., 2016b), we argue that the success of any of these methods may depend more on the specific understanding and operation leading to internal reproducibility of each individual technique's results than an inherent superiority of one technique over another."

We will include the amount of sample material used per laboratory in Table 1. We will further include the following in the results section: "In relation to the amount of used sample material, most labs either introduced 10 or 20 g to their system no matter the extraction approach (I or II), soil type or WC. Only labs 11 and 13 chose different weights with respect to the WC, e.g., 10 g for the higher WC (20%) and 20 g for 8% WC for extraction approach I."

**References**

Millar, C., Pratt, D., Schneider, D. J. and McDonnell, J. J.: A comparison of extraction systems for plant water stable isotope analysis, Rapid Commun. Mass Spectrom., 32(13), 1031–1044, doi:10.1002/rcm.8136, 2018.

Orlowski, N., Pratt, D. L. and McDonnell, J. J.: Intercomparison of soil pore water extraction methods for stable isotope analysis, Hydrol. Process., 30(19), 3434–3449, doi:10.1002/hyp.10870, 2016b.

Orlowski, N., Winkler, A., McDonnell, J. J. and Breuer, L.: A simple greenhouse experiment to explore the effect of cryogenic water extraction for tracing plant source water, Ecohydrology, e1967, doi:10.1002/eco.1967, 2018.

---

## Author Comment (AC3) · 30 May 2018

**Reply to Reviewer 3**

In the following please find the corrections and comments to the referee's response.

**General comments**

This manuscript reports the results of an intercomparison exercise that aimed at testing the consistency of cryogenic water extractions for the analysis of stable isotope of hydrogen and oxygen in soil water among worldwide-distributed laboratories. In the last few years, the ecohydrological and soil science international communities have shown a strong and increasing interest in better understanding the functional interrelationships between soil and vegetation based on the use of stable water isotope data. The cryogenic water extraction technique has so become a golden standard for sampling water from the unsaturated zone. Very recent studies, often conducted by the first author of this manuscript and colleagues, already showed some potentials and limitations of this technique, and provided helpful information for users. However, a worldwide interlaboratory comparison among several cryogenic extraction facilities was still missing, thus this work it is certainly welcome. Indeed, I believe that this manuscript is timely and of great interest for the readers of this journal. The manuscript is very well written, logically structured, nicely illustrated, and the conceptual steps can be followed very well. The working hypothesis and the specific objectives are well posed, following a substantial introduction, the statistical analysis is correct, and the results and interpretation are well supported by the data. The methodological approach leading to the comparison exercise was solidly defined and clearly presented. As noted by the Authors themselves, the large differences in performance found among the labs included in the exercise are somehow worrisome and pose questions on the possible adoption of cryogenic water extraction as a standard method for soil water sampling. However, these results are very relevant to the scientific community because implicitly suggest cautions in comparing isotope soil water data extracted by different facilities, and indicate that much technical work is still needed to test possible further controls on these differences and develop new techniques able to return more consistent results.

We thank Daniele Penna for taking the time to review our manuscript and providing this very positive feedback.

**Specific comments**

1. P4 L11. Here, and throughout the rest of the manuscript (e.g., P7 L16; P9 L12-14; P11 L15. . .), it is not immediately clear what "isotope results" are, and I suggest to replace this term with "values" or "data".

**Response:** We will change it to "data" or "values" or "composition" as suggested.

2. P5 L14. The authors reported that the soil was homogenized before shipping. However, as noted by the other reviewers, I wonder if possible heterogeneities in the analysed soil samples (especially for small volumes) could have been present and could have affected the results.

**Response:** For our intercomparison, we used "standard lab soils" from the LUFA Speyer (https://www.lufa-speyer.de/index.php/dienstleistungen/standardboeden/8-dienstleistungen/artikel/57-standard-soils). In guidelines of the German JKI (Julius-Kuehn-Institute) and other relating guidelines (OECD), soils with certain characteristics are recommended for such studies. The LUFA lab already provided dried, homogenized and sieved soils (with a 2 mm screen)

according to "Good Laboratory Practice". However, we repeated the homogenization and oven-drying process to ensure that all subsamples that we shipped to the respective labs were equal.

3. P5 L27. I find the definition of "water-water" cryogenic extraction a bit confusing. I suggest to use, throughout the manuscript, simply the terms "water extraction" vs. "soil water extraction", or something similar.

**Response:** We will follow the reviewer's suggestion.

4. P7 L1-2. Although intuitive, I suggest to add a short explanation about the choice of applying different extraction times for the silty sand and the clayey loam soils.

**Response:** We will add the following to the revised version of the manuscript: "Since different extraction times and temperatures were applied in past studies, we decided that participating laboratories should follow two different extraction approaches..." and further "For comparison, in past studies extraction times from 2.5 min (Koeniger et al., 2011) over 30 min (West et al., 2006) to 7 h (Araguás-Araguás et al., 1995) for sandy soils and from 30 min (Goebel & Lascano, 2012) over 40 min (West et al., 2006) to 8 h (Araguás-Araguás et al., 1995) for clayey soils were reported."

5. P8 L28, and P12 L6-20, and Figs. 5-6. It seems to me that these are not "true evaporation lines" describing the progressive isotopic enrichment of an individual source water (see Benettin et al., 2018 who reported the often misused term and concept of evaporation line). This might not be a critical point in the interpretation of the results and the overall meaning of the research. However, for the sake of accuracy, I suggest to check this and in case change the terminology (eg, simply calling them regression lines) and slightly re-interpret the results reported at P12 L6-20. Moreover, it's not very clear to me why in the left panels of Figs. 5 and 6 (8% WC) one regression line (for 8% samples) is reported in addition to the GMWL whereas in right panels of Figs. 5 and 6 (20% WC) two regression lines are shown (both for 8% and 20% WC). Please, fix this or explain.

**Response:** We will change the term to "regression lines". We will further add the following to the respective section on P12: "Benettin et al. (2018) recently revised the widely used concept of evaporation lines. The authors question that the trend line passing through fractionated soil water samples correctly identifies their source water and emphasis that trend lines through evaporated samples can differ widely from true evaporation lines."
For clarity, we deleted regression lines for the 8% WC versions from the right panels of Figures 5 and 6.

6. P9 L23-24. Do the Authors have any idea about the reason for recovery rates higher than 100%? Could this somehow affect the results? Perhaps a sentence could be added here (trying to avoid the risk of speculation).

**Response:** We already included a potential explanation for this phenomenon in the discussion section of our previous manuscript version: "This could be attributed either to leaky vacuum systems (which might allow atmospheric water vapor to enter the system) or to a remoistening of the oven-dried soil samples before water extraction."

7. P11 L27. In addition to the statistical results, I wonder whether it might be appropriate to show OA-ICOS and IRMS data as boxplots to graphically stress the difference between values returned by the two techniques.

**Response:** Figures 3 and 4 show the mean of the three replicates with respect to the reference DI water and y-error bars stand for the isotopic variation of the replicates. Since our sample size (n=3) is relatively small, we decided to display only the data's central tendency by using traditional mean-and-error scatter plots. The decision follows recommendations by Nature methods (2014) and we hope that the reviewer will concur with this assessment (a great suggestion though, if we had a larger n).

8. P17 L23. The reference of "Orlowsky et al., 2018" is missing from the reference list.

**Response:** We will add the missing reference.

9. Fig. 1. I suggest to increase the size of the axis labels.

**Response:** We will follow the reviewer's suggestion.

10. Fig. 3 and Fig. 4. In the caption: was the mean computed among the three replicates? If so, I suggest to specify this for the sake of clarity.

**Response:** We will edit the sentence as follows: "Symbols represent the mean of the three replicates and y-error bars stand for the isotopic variation of the replicates."

11. Fig. 5 and Fig. 6. I suggest to add in the caption that the legend includes explanation for the symbols used for the 16 labs and the two extraction approaches.

**Response:** We will add the following: "…for OA-ICOS and IRMS data (upper and lower panels, respectively) from the 16 participating labs (different colors represent different labs) and both extraction methods (lab-procedure: I and pre-defined: II). For reference, plots include the Global Meteoric Water Line (GMWL, solid red line) and soil water regression lines for 8% and 20% WC (solid green and orange lines, respectively)."

**References**

Araguás-Araguás, L., Rozanski, K., Gonfiantini, R., & Louvat, D. (1995). Isotope effects accompanying vacuum extraction of soil water for stable isotope analyses. *Journal of Hydrology*, *168*(1–4), 159–171. https://doi.org/10.1016/0022-1694(94)02636-P

Benettin, P., Volkmann, T. H. M., von Freyberg, J., Frentress, J., Penna, D., Dawson, T. E., & Kirchner, J. W. (2018). Effects of climatic seasonality on the isotopic composition of evaporating soil waters. *Hydrol. Earth Syst. Sci.*, *22*(5), 2881–2890. https://doi.org/10.5194/hess-22-2881-2018

Goebel, T. S., & Lascano, R. J. (2012). System for high throughput water extraction from soil material for stable isotope analysis of water. *Journal of Analytical Sciences, Methods and Instrumentation*, *02*(04), 203–207. https://doi.org/10.4236/jasmi.2012.24031

Koeniger, P., Marshall, J. D., Link, T., & Mulch, A. (2011). An inexpensive, fast, and reliable method for vacuum extraction of soil and plant water for stable isotope analyses by mass spectrometry. *Rapid Communications in Mass Spectrometry*, *25*(20), 3041–3048. https://doi.org/10.1002/rcm.5198

Nature methods. (2014, February). Visualizing samples with box plots. Retrieved May 22, 2018, from https://www.nature.com/nmeth/volumes/11/issues/2

West, A. G., Patrickson, S. J., & Ehleringer, J. R. (2006). Water extraction times for plant and soil materials used in stable isotope analysis. *Rapid Communications in Mass Spectrometry*, *20*(8), 1317–1321. https://doi.org/10.1002/rcm.2456